# Fork-Merge Decoding: Enhancing Multimodal Understanding in Audio-Visual Large Language Models

## Abstract

The goal of this work is to enhance balanced multimodal understanding in audio-visual large language models (AV-LLMs) by addressing modality bias without additional training. In current AV-LLMs, audio and video features are typically processed jointly in the decoder. While this strategy facilitates unified multimodal understanding, it may introduce modality bias, where the model tends to over-rely on one modality due to imbalanced training signals. To mitigate this, we propose Fork-Merge Decoding (FMD), a simple yet effective inference-time strategy that requires no additional training or architectural modifications. FMD first performs modality-specific reasoning by processing audio-only and video-only inputs through the early decoder layers (*fork*), and then merges the resulting hidden states for joint reasoning in the remaining layers (*merge*). This separation allows each modality to be emphasized in the early stages while encouraging balanced contributions during integration. We validate our method on three representative AV-LLMs—VideoLLaMA2, video-SALMONN, and Qwen2.5-Omni—using three benchmark datasets. Experimental results show consistent gains in audio, video, and audio-visual reasoning tasks, highlighting the effectiveness of inference-time interventions for robust and efficient multimodal understanding.

## 1 Introduction

Recent advancements in large language models (LLMs) have demonstrated superior performance in various text-centric tasks such as problem solving, translation, and summarization (Achiam et al., 2023; Brown et al., 2020; Liu et al., 2023b; Thoppilan et al., 2022; Wei et al., 2022a;b; Zhao et al., 2023a). Building on these successes in text processing, LLMs have evolved to handle additional modalities, including images (Alayrac et al., 2022; Chen et al., 2023a; Dai et al., 2023; Huang et al., 2023; Li et al., 2023a; Liu et al., 2023a; Yu et al., 2024; Zhang et al., 2024; Zhu et al., 2023b), videos (Lin et al., 2024; Maaz et al., 2024), and audio (Huang et al., 2024; Rubenstein et al., 2023; Tang et al., 2024), giving rise to multimodal LLMs (MLLMs). These models process diverse modalities through separate encoders and integrate their outputs within a decoder language model, achieving remarkable performance across a wide range of tasks. Among these, audio-visual LLMs (AV-LLMs) are particularly notable for their ability to jointly integrate visual and auditory information, supporting more sophisticated reasoning and achieving closer alignment with human multimodal perception (Cheng et al., 2024; Sun et al., 2024; Zhang et al., 2023; Xu et al., 2025).

To effectively leverage pretrained LLM decoders in AV-LLMs, various fusion strategies have been proposed to integrate audio and visual information. One common approach (Chen et al., 2023b; Cheng et al., 2024; Chowdhury et al., 2024; Han et al., 2024; Lyu et al., 2023; Panagopoulou et al., 2023; Ye et al., 2024; Zhan et al., 2024; Zhang et al., 2023; Zhao et al., 2023b; Xu et al., 2025) is token-wise fusion, where audio and visual features are extracted by separate encoders and then concatenated along the sequence dimension before being fed into the decoder as a continuous input sequence. Several studies (Chowdhury et al., 2024; Ye et al., 2024) additionally introduce adapter modules that facilitate interaction between audio and visual features before they are passed into the LLM. Another approach (Han et al., 2023; Su et al., 2023; Sun et al., 2024) is channel-wise fusion, in which modality-specific features are concatenated along the channel dimension to form a unified representation. In most current AV-LLM architectures, the decoder receives both audio and visual

inputs simultaneously, which raises a potential concern: if the model finds one modality easier to interpret—perhaps due to better alignment with its pretraining objectives—it may over-rely on that modality, leading to modality bias and modality-specific hallucinations (Leng et al., 2024a).

To investigate this possibility, we begin by analyzing the attention weight distributions over audio-visual inputs using 100 samples from the AVHBench dataset (Sung-Bin et al., 2025). Our analysis examines the final decoder layer of VideoLLaMA2 (Cheng et al., 2024), focusing on the attention weights of the last token. Since the token is critical for predicting the next token, we use it to quantify the relative attention allocated to each modality. As shown in Figure 1, the vanilla decoding setup, which reflects the default inference behavior of the model, exhibits a clear bias toward video inputs, with attention disproportionately concentrated on visual features over audio. This observation aligns with findings from recent studies (Guan et al., 2024; Leng et al., 2024a; Nishimura et al., 2024; Wang et al., 2024), which report that MLLM decoders often exhibit modality bias.

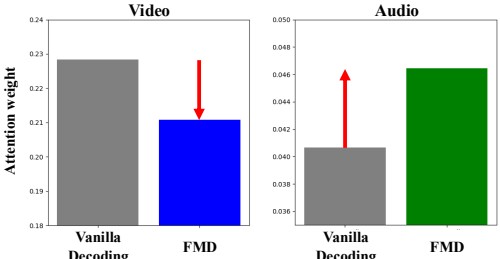

Figure 1: **Attention weight analysis in VideoL-LaMA2 on the AVHBench dataset.** We analyze 100 samples and examine the attention weights from the last decoder layer, focusing on the final token of the question. Attention is disproportionately allocated to video inputs over audio, revealing a modality bias. Our proposed FMD method reduces this gap by encouraging more balanced contributions from both modalities.

These studies highlight that such imbalances can lead to hallucinations or flawed reasoning, emphasizing the need to mitigate modality bias for more balanced multimodal understanding.

To address this issue, we propose Fork-Merge Decoding (FMD), which is a simple yet effective strategy that enhances multimodal understanding without altering the AV-LLM architecture or requiring additional training. The core idea of FMD is to divide the decoding phase into two stages: a *fork* phase and a *merge* phase, designed to improve both unimodal and multimodal understanding. In the *fork* phase, the original multimodal input is split into two unimodal branches by zeroing out either the visual or auditory modality while retaining the text question. Each branch is processed independently through the initial layers of the pretrained AV-LLM, producing modality-specific hidden representations without requiring additional full forward passes. In the *merge* phase, these representations are combined and passed through the remaining decoder layers. This separation enables the model to first attend to unimodal cues in isolation before integrating them for complementary multimodal understanding. As shown in Figure 1, FMD reduces the attention weight on video inputs by 14% and increase the weight on audio inputs by 7%. This adjustment balances the modality bias while still preserving the attributes of the pretrained model. Building on this, since recent AV-LLMs commonly adopt either token-wise or channel-wise concatenation for audio-visual fusion, we propose a generalized decoding strategy that is compatible with both fusion methods. This unified approach improves performance across a variety of models, regardless of their fusion mechanism.

Furthermore, we evaluate the effectiveness of FMD by applying it to three recent AV-LLMs: VideoLLaMA2 (Cheng et al., 2024) and Qwen2.5-Omni (Xu et al., 2025) (token-wise fusion) and video-SALMONN (Sun et al., 2024) (channel-wise fusion). Applying FMD leads to consistent performance improvements in all baselines across three widely used audio-visual benchmarks: AVQA (Yang et al., 2022), MUSIC-AVQA (Li et al., 2022), and AVHBench (Sung-Bin et al., 2025). Notably, FMD enhances performance not only in tasks that emphasize a single modality but also in tasks that require balanced reasoning across both modalities. These results show that FMD enhances multimodal understanding by fully utilizing information from each modality during inference.

## 2 PRELIMINARIES

**Input processing.** Most existing AV-LLMs (Chen et al., 2023b; Cheng et al., 2024; Chowdhury et al., 2024; Lyu et al., 2023; Panagopoulou et al., 2023; Sun et al., 2024; Ye et al., 2024; Zhan et al., 2024; Zhang et al., 2023; Zhao et al., 2023b) process audio and video separately before feeding them into an LLM decoder. Visual inputs are encoded frame by frame into spatial embeddings, while audio signals are mapped to semantic representations capturing acoustic and prosodic cues. Textual instructions or questions are tokenized and embedded by a tokenizer of language model.

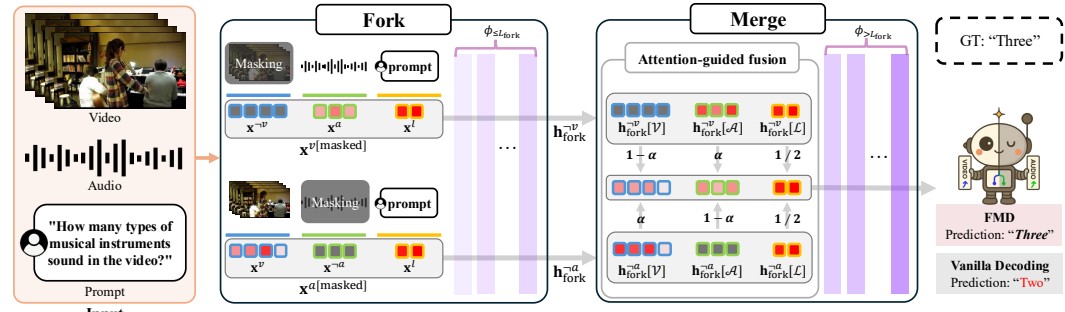

Figure 2: **Overview of the Fork-Merge Decoding pipeline.** The AV-LLM takes video frames, an audio waveform, and a question prompt as input. In the *fork* phase, FMD masks one modality while preserving the question, enabling independent reasoning. After $L_{\text{fork}}$ decoder layers, the *merge* phase combines $\boldsymbol{h}_{\text{fork}}^{\neg v}$ and $\boldsymbol{h}_{\text{fork}}^{\neg a}$ with an attention-derived weight $\alpha$, and the merged representation is processed by the remaining layers to generate answers with balanced multimodal understanding.

For token-wise fusion models, video frames are transformed into $M$ embeddings $\boldsymbol{x}^v = \{\boldsymbol{x}_1, \ldots, \boldsymbol{x}_M\}$ and audio into $N$ embeddings $\boldsymbol{x}^a = \{\boldsymbol{x}_{M+1}, \ldots, \boldsymbol{x}_{M+N}\}$. These are concatenated with $L$ text embeddings $\boldsymbol{x}^l = \{\boldsymbol{x}_{M+N+1}, \ldots, \boldsymbol{x}_{M+N+L}\}$ to form the final sequence $\boldsymbol{x} = \boldsymbol{x}^v \oplus \boldsymbol{x}^a \oplus \boldsymbol{x}^l$ of length $M + N + L$, omitting instruction tokens for simplicity. For channel-wise fusion models, visual and audio features are projected to a fixed length $U$ embeddings and then concatenated along the channel dimension to form joint audio-visual embeddings, $\boldsymbol{x}_i^{av} = [\boldsymbol{x}_i^v; \boldsymbol{x}_i^a]$. The resulting set $\boldsymbol{x}_i^{av} = \{\boldsymbol{x}_1^{av}, \ldots, \boldsymbol{x}_U^{av}\}$ is combined with $L$ text embeddings via token-wise concatenation, $\boldsymbol{x} = \{\boldsymbol{x}_1^{av}, \ldots, \boldsymbol{x}_U^{av}, \boldsymbol{x}_{U+1}^l, \ldots, \boldsymbol{x}_{U+L}^l\}$, which is then fed into the decoder.

**Decoding.** Both token-wise and channel-wise fusion-based AV-LLMs generate outputs from the input sequence $\boldsymbol{x}$ using an autoregressive decoding strategy, where each token is predicted by attending to previously generated tokens under a causal mask. At each decoding step $t$, the model generates the next token $\boldsymbol{y}_t$ conditioned on the input sequence $\boldsymbol{x}$, which includes video, audio, and text prompts, as well as the previously generated tokens $\boldsymbol{y}_{<t}$, with $\boldsymbol{y}_t \sim p(\boldsymbol{y}_t | \boldsymbol{x}, \boldsymbol{y}_{<t}) \propto \exp\left(\text{logit}(\boldsymbol{y}_t | \boldsymbol{x}, \boldsymbol{y}_{<t})\right)$.

# 3 FORK-MERGE DECODING FOR AUDIO-VISUAL UNDERSTANDING

This section introduces Fork-Merge Decoding (FMD) for *token-wise* (Section 3.1) and *channel-wise* (Section 3.2) fusion in AV-LLMs. An overview of the process is illustrated in Figure 2.

## 3.1 DECODING WITH TOKEN-WISE FUSION IN AV-LLMs

**Input masking strategy.** In token-wise fusion models like VideoLLaMA2 (Cheng et al., 2024), the input sequence is composed of $\boldsymbol{x} = \boldsymbol{x}^v \oplus \boldsymbol{x}^a \oplus \boldsymbol{x}^l$, where $\boldsymbol{x}^v$, $\boldsymbol{x}^a$, and $\boldsymbol{x}^l$ denote visual, audio, and language embeddings, respectively. To enable modality-specific processing for audio and video while preserving the textual question, we create two masked input variants as follows:

$$\boldsymbol{x}^{v[\text{masked}]} = \boldsymbol{x}^{\neg v} \oplus \boldsymbol{x}^a \oplus \boldsymbol{x}^l, \quad \boldsymbol{x}^{a[\text{masked}]} = \boldsymbol{x}^v \oplus \boldsymbol{x}^{\neg a} \oplus \boldsymbol{x}^l, \tag{1}$$

where $\boldsymbol{x}^{\neg v}$ and $\boldsymbol{x}^{\neg a}$ denote the modality-masked embeddings for vision and audio, respectively. These are obtained by zeroing out the corresponding video frames or audio waveforms at the input level. This preserves original embedding shapes and positions while removing content information.

**Fork processing.** To encourage independent understanding over each modality, each masked sequence is separately processed through the first $L_{\text{fork}}$ transformer layers of the decoder $\phi$:

$$\boldsymbol{h}_{\text{fork}}^{\neg v} = \phi_{\leq L_{\text{fork}}}(\boldsymbol{x}^{v[\text{masked}]}), \quad \boldsymbol{h}_{\text{fork}}^{\neg a} = \phi_{\leq L_{\text{fork}}}(\boldsymbol{x}^{a[\text{masked}]}), \tag{2}$$

where $\boldsymbol{h}_{\text{fork}}^{\neg v}$ and $\boldsymbol{h}_{\text{fork}}^{\neg a}$ denote the intermediate hidden states obtained from the vision-masked and audio-masked inputs, respectively. This design ensures that the model does not observe both audio and visual inputs simultaneously in the early stages, allowing it to focus on how each modality

individually relates to the textual prompt. By reasoning over each modality in isolation, the model is less likely to become biased toward the more dominant or easier-to-interpret modality.

**Merge processing.** After the $L_{\text{fork}}$ layers, the hidden states $\boldsymbol{h}_{\text{fork}}^{\neg v}$ and $\boldsymbol{h}_{\text{fork}}^{\neg a}$ are fused and passed through the remaining transformer layers. Each hidden state has a sequence length of $M + N + L$, corresponding to the visual, audio, and text tokens. The fusion is performed by summing the corresponding embeddings at each modality position, and is formally defined as follows:

$$
\begin{aligned}
\boldsymbol{h}_{\text{merge}}[\mathcal{V}] &= (1 - \alpha) \cdot \boldsymbol{h}_{\text{fork}}^{\neg v}[\mathcal{V}] + \alpha \cdot \boldsymbol{h}_{\text{fork}}^{\neg a}[\mathcal{V}], \\
\boldsymbol{h}_{\text{merge}}[\mathcal{A}] &= \alpha \cdot \boldsymbol{h}_{\text{fork}}^{\neg v}[\mathcal{A}] + (1 - \alpha) \cdot \boldsymbol{h}_{\text{fork}}^{\neg a}[\mathcal{A}], \\
\boldsymbol{h}_{\text{merge}}[\mathcal{L}] &= \tfrac{1}{2} \left( \boldsymbol{h}_{\text{fork}}^{\neg v}[\mathcal{L}] + \boldsymbol{h}_{\text{fork}}^{\neg a}[\mathcal{L}] \right),
\end{aligned}
\tag{3}
$$

where $\mathcal{V} = [1{:}M]$, $\mathcal{A} = [M{+}1{:}M{+}N]$, and $\mathcal{L} = [M{+}N{+}1{:}M{+}N{+}L]$ denote the index ranges corresponding to the visual, audio, and language embeddings, respectively. The final merged representation $\boldsymbol{h}_{\text{merge}}$ is then constructed by concatenating the modality-specific segments along the token dimension as $\boldsymbol{h}_{\text{merge}} = \boldsymbol{h}_{\text{merge}}[\mathcal{V}] \oplus \boldsymbol{h}_{\text{merge}}[\mathcal{A}] \oplus \boldsymbol{h}_{\text{merge}}[\mathcal{L}]$. Here, $\alpha$ is a fusion weight for unmasked segments, reflecting their relative contribution when combining masked and unmasked segments. We refer to this approach as attention-guided fusion and describe in detail in the following section.

**Attention-guided fusion.** To determine the fusion weight $\alpha$, we use the attention matrix $\mathbf{A}^{\text{final}} \in \mathbb{R}^{T \times T}$ from the final transformer layer. Specifically, we focus on the attention vector of the last token, $\mathbf{a}_{\text{last}} = \mathbf{A}_{T,:}^{\text{final}} \in \mathbb{R}^T$, which is critical in next-token prediction and follows the approach in prior studies (Huo et al., 2025; Song et al., 2024). By leveraging the two masked branches $\boldsymbol{h}_{\text{fork}}^{\neg v}$ and $\boldsymbol{h}_{\text{fork}}^{\neg a}$, we compute the attention-based contributions of unmasked segments by summing the attention mass over the corresponding regions relative to the total mass, which are then used as $\alpha$:

$$
\alpha = \frac{\sum_{i \in \mathcal{A}} \mathbf{a}_{\text{last}}^{v[\text{masked}]}[i] + \sum_{i \in \mathcal{V}} \mathbf{a}_{\text{last}}^{a[\text{masked}]}[i]}{\sum_{i \in \mathcal{V} \cup \mathcal{A}} \left( \mathbf{a}_{\text{last}}^{v[\text{masked}]}[i] + \mathbf{a}_{\text{last}}^{a[\text{masked}]}[i] \right)},
\tag{4}
$$

where $\mathbf{a}_{\text{last}}^{v[\text{masked}]}$ and $\mathbf{a}_{\text{last}}^{a[\text{masked}]}$ represent the attention weight distributions of the final token from $\boldsymbol{h}_{\text{fork}}^{\neg v}$ and $\boldsymbol{h}_{\text{fork}}^{\neg a}$, respectively. The final attention weight $\alpha$ is then obtained as the fraction of unmasked attention over the total attention in Eq. 4 and is used to interpolate between $\boldsymbol{h}_{\text{fork}}^{\neg v}$ and $\boldsymbol{h}_{\text{fork}}^{\neg a}$ in Eq. 3. In practice, instead of computing $\alpha$ for each data point, we estimate a representative value by sampling 100 random examples from the AVHBench (Sung-Bin et al., 2025) dataset. This approach is motivated by two considerations: (1) computing a separate $\alpha$ for each sample increases inference time, as it requires two additional full forward passes to obtain $\mathbf{a}_{\text{last}}^{v[\text{masked}]}$ and $\mathbf{a}_{\text{last}}^{a[\text{masked}]}$, (2) noisy sample-specific $\alpha$ values can act as outliers, causing performance drops, as shown by the results labeled Attention (Adaptive) in Table 3. The resulting representative $\alpha$ is then applied consistently across all experiments, including datasets beyond AVHBench, to verify its generalizability.

This attention-guided fusion ensures that structurally aligned hidden states are preserved, while allowing the more informative modality to be emphasized. It enables a flexible and interpretable merging scheme without disrupting the architectural integrity of pretrained AV-LLMs.

**Decoding.** The merged hidden state is then forwarded through the remaining transformer layers: $\boldsymbol{h}_{\text{final}} = \phi_{> L_{\text{fork}}}(\boldsymbol{h}_{\text{merge}})$, producing the final prediction logits. By delaying modality fusion to deeper layers, where individual representations become semantically richer through the *fork* phase, our method enhances multimodal understanding while mitigating issues caused by modality imbalance.

### 3.2 Decoding with channel-wise fusion in AV-LLMs

**Input masking strategy.** In channel-wise fusion models such as video-SALMONN (Sun et al., 2024), the input sequence is structured as $\boldsymbol{x} = \{\boldsymbol{x}_1^{av}, \ldots, \boldsymbol{x}_U^{av}, \boldsymbol{x}_{U+1}^l, \ldots, \boldsymbol{x}_{U+L}^l\}$, where $\boldsymbol{x}^{av}$ denotes audio-visual embeddings and $\boldsymbol{x}^l$ corresponds to language (prompt) embeddings. Here, $U$ and $L$ indicate the number of audio-visual and language sequence elements, respectively. To allow for modality-specific processing, we construct two masked variants of the input:

$$
\boldsymbol{x}^{v[\text{masked}]} = \boldsymbol{x}^{a \neg v} \oplus \boldsymbol{x}^l, \quad \boldsymbol{x}^{a[\text{masked}]} = \boldsymbol{x}^{\neg av} \oplus \boldsymbol{x}^l,
\tag{5}
$$

where $\boldsymbol{x}^{a \neg v}$ and $\boldsymbol{x}^{\neg av}$ denote the video-masked and audio-masked embeddings, respectively, obtained by zeroing out the corresponding inputs before they are passed into the decoder layers.

**Fork-merge decoding.** $h_{\text{fork}}^{\neg v}$ and $h_{\text{fork}}^{\neg a}$ are obtained by passing the masked inputs $x^{v[\text{masked}]}$ and $x^{a[\text{masked}]}$ through the $L_{fork}$ layers (refer to Eq. 2). To compute the merged hidden state $h_{\text{merge}}$, we perform element-wise addition over the audio-visual embedding representations from both branches, based on the assumption that they capture complementary information. Since each branch processes modality-masked inputs (i.e., audio-masked for visual features and vice versa), their combination is expected to yield a more complete representation. Additionally, mean pooling is applied over the question prompt embedding positions to maintain consistency:

$$h_{\text{merge}}[i] = \begin{cases} h_{\text{fork}}^{\neg v}[i] + h_{\text{fork}}^{\neg a}[i], & \text{if } i \le U, \\ \frac{1}{2}\left(h_{\text{fork}}^{\neg v}[i] + h_{\text{fork}}^{\neg a}[i]\right), & \text{if } i > U. \end{cases} \tag{6}$$

We do not apply attention-guided fusion in the channel-wise setting, as the hidden states do not disentangle audio and visual embeddings along time axis. Subsequently, decoding is then continued by forwarding the merged hidden state $h_{\text{merge}}$ through the remaining decoder layers to produce the final output logits. Notably, FMD provides a unified framework that seamlessly integrates input masking with modality-specific processing in separate decoder branches and merged decoding, making it broadly applicable to a wide range of AV-LLM architectures, regardless of their fusion strategies.

## 4 EXPERIMENTS

### 4.1 EXPERIMENTAL SETUP

**Baselines.** We evaluate our approach using three representative AV-LLMs: VideoLLaMA2 (Cheng et al., 2024), video-SALMONN (Sun et al., 2024), and Qwen2.5-Omni (Xu et al., 2025).

**Datasets and evaluation protocol.** The AVQA (Yang et al., 2022) dataset contains 57,000 YouTube videos for evaluating real-world audio-visual understanding. MUSIC-AVQA (Li et al., 2022) offers 45,867 QA pairs from 9,288 music performance videos, focusing on fine-grained audio-visual reasoning such as identifying sound sources and temporally aligning auditory and visual cues. AVH-Bench (Sung-Bin et al., 2025) is the first benchmark specifically designed to assess audio-visual hallucinations in AV-LLMs. It comprises four subtasks: audio-driven video hallucination (A→V), video-driven audio hallucination (V→A), audio-visual matching (AV matching), and audio-visual captioning (AV captioning). AVUT (Yang et al., 2025) and WorldSense (Hong et al., 2025) datasets evaluate video comprehension with an emphasis on auditory cues and general multimodal video understanding, respectively. As VideoLLaMA2 use fixed sparse sampling (e.g., 8 frames), we filter AVUT and WorldSense to include only videos under 60 seconds to preserve information density. For the transformer layer analysis in Section 4.4, we select 200 samples from each task (A→V, V→A, and AV matching), with the remaining data used for evaluation. For the AVHBench dataset, the three binary (yes/no) tasks except AV captioning, we report classification accuracy. For AVQA, MUSIC-AVQA, AVUT, WorldSense and AV captioning, which involve open-ended responses, we follow the GPT-assisted evaluation protocol from the official VideoLLaMA2 implementation[1].

**Implementation details.** For attention-guided fusion, we set the weighting parameter $\alpha$ as described in Section 3.1, using 100 randomly sampled examples from the AVHBench dataset: $\alpha = 0.8$ for VideoLLaMA2 and $\alpha = 0.9$ for Qwen2.5-Omni. The number of layers used for the *fork* phase $L_{fork}$ is chosen to be roughly one-seventh of the total decoding layers: the 4th layer for VideoLLaMA2 and Qwen2.5-Omni (28 layers) and the 6th layer for video-SALMONN (40 layers). The rationale for these *fork* layer selections is further analyzed in Section 4.4.

### 4.2 QUANTITATIVE ANALYSIS

**Comparison with vanilla decoding.** To verify the effectiveness of our proposed FMD method, we evaluate it with VideoLLaMA2, video-SALMONN, and Qwen2.5-Omni on five datasets: AVQA, MUSIC-AVQA, AVHBench, AVUT, and WorldSense. As shown in Table 1, applying FMD consistently improves performance over vanilla decoding, which represents the original model inference, across all tasks. Notably, the gains are more pronounced in the AV captioning task (evaluated on a 5-point scale), which requires generating long and sophisticated answers, with relative improvements ranging from 3.4% to 9.8% across models. Among the models, video-SALMONN benefits the most, achieving a remarkable 12.26% increase on the AVQA dataset. This is particularly signif-

---

[1] https://github.com/DAMO-NLP-SG/VideoLLaMA2/tree/audio_visual

Table 1: **Comparison of audio-visual understanding performance.** We evaluate FMD on Video-oLLaMA2, video-SALMONN, and Qwen2.5-Omni. *Vanilla* denotes the original decoding strategy of each model. FMD consistently improves performance across all benchmarks, especially in AV matching (AV mat.) and AV captioning (AV cap.). The entries for Qwen2.5-Omni on the MUSIC-AVQA, AVUT, and WorldSense benchmarks are marked as N/A due to out-of-memory constraints.

| Model | Decoding | AVHBench | | | | AVQA | MUSIC-AVQA | AVUT | WorldSense |
|---|---|---|---|---|---|---|---|---|---|
| | | A→V | V→A | AV Mat. | AV Cap. | | | | |
| VideoLLaMA2 | Vanilla | 80.02 | 77.03 | 57.75 | 2.84 | 60.23 | 81.30 | 44.44 | 29.41 |
| | **FMD** | **80.45** | **77.52** | **59.01** | **2.95** | **61.46** | **81.50** | **45.13** | **29.91** |
| video-SALMONN | Vanilla | 68.69 | 62.39 | 49.46 | 1.83 | 28.20 | 44.48 | 32.36 | 56.20 |
| | **FMD** | **70.51** | **65.41** | **54.77** | **2.01** | **40.46** | **50.60** | **32.85** | **57.83** |
| Qwen2.5-Omni | Vanilla | 80.77 | 71.20 | 77.45 | 3.25 | 86.49 | N/A | N/A | N/A |
| | **FMD** | **81.30** | **71.77** | **78.22** | **3.36** | **86.61** | N/A | N/A | N/A |

icant because video-SALMONN has not been trained on AVQA or MUSIC-AVQA, yet FMD still enhances its zero-shot inference. These results highlight the robustness and strong generalizability of our approach. We further validate the efficacy of FMD by applying it to the more recent AV-LLM, video-SALMONN 2+ (Tang et al., 2025) in Supp. A.2.

**Comparison with other decoding methods.** We compare our FMD method with other test-time decoding strategies that operate at the logit level. The evaluation is conducted using the VideoLLaMA2 model on the AVHBench dataset. Specifically, the comparison includes the following methods: DoLa (Chuang et al., 2024), VCD (Leng et al., 2024b), SID (Huo et al., 2025), and FMD variants with Gaussian noise injection and zero-out masking. **DoLa** contrasts intermediate-layer outputs with final predictions to factually correct outputs. **VCD** reduces language bias by injecting Gaussian noise into the visual input and subtracting the resulting logits from the original ones. **SID**

Table 2: **Comparison of decoding methods on AVHBench dataset using VideoLLaMA2.** We compare vanilla decoding with DoLa, VCD, SID and two FMD variants (Gaussian noise injection and zero-out masking). Among them, FMD with zero-out masking achieves the highest overall accuracy, underscoring its effectiveness.

| Decoding | Designed for | AVHBench | | |
|---|---|---|---|---|
| | | A→V | V→A | AV Matching |
| Vanilla | - | 80.02 | 77.03 | 57.75 |
| DoLa (Chuang et al., 2024) | LLM | 69.34 | 63.44 | 48.33 |
| VCD (Leng et al., 2024b) | VLM | 75.96 | 69.67 | 52.52 |
| SID (Huo et al., 2025) | VLM | 78.53 | 72.82 | 53.52 |
| FMD w/ noise | AV-LLM | 79.17 | **78.03** | 57.76 |
| **FMD w/ zero-out** | AV-LLM | **80.45** | 77.52 | **59.01** |

adopts a similar approach with VCD but preserves the least informative visual tokens based on attention weights, contrasting their logits with those of the original outputs. For DoLa, we follow the original setup and extract outputs from the same intermediate layer reported in the paper. For VCD and SID, which were originally developed for vision-language models (VLMs), we adapt their procedures to handle both audio and video modalities in AV-LLMs.

As shown in Table 2, applying decoding strategies developed for LLMs or VLMs to AV-LLMs leads to degraded performance. This highlights the need for decoding methods that are specifically tailored to the unique characteristics of AV-LLMs, which differ from those of unimodal or bimodal models. Additionally, injecting Gaussian noise into the inputs within FMD (denoted as FMD with noise) results in lower performance on A→V and AV matching tasks compared to vanilla decoding, although it does lead to improved performance on the V→A task. We attribute this to the inability of Gaussian noise to equally isolate modality-specific information, as further discussed in Supp. A.3. In contrast, zero-out masking within FMD yields consistent performance improvements across all tasks, demonstrating that the proposed FMD design is suitable for audio-visual understanding.

### 4.3 QUALITATIVE ANALYSIS

**Audio-visual matching from AVHBench (Figure 3a).** The video shows a man fishing in a stream, and the audio contains the sound of water flowing in a valley with occasional splashes from footsteps. The vanilla decoding mainly captures the visual content, describing only the presence of a man fishing in a stream. However, it overlooks the acoustic context, failing to reflect the sound of flowing water and splashes. By contrast, our proposed FMD generates a more comprehensive description that integrates both modalities, enriching the visual detail (e.g., specifying the fishing rod) and the audio detail (e.g., capturing the sounds of water flow and splashes). This highlights the strength of FMD in balancing and fusing multimodal cues, leading to richer and more faithful outputs.

**Audio-visual question & answering from AVQA (Figure 3b).** The video shows a train, but in the early frames its appearance could be confused with a cable car. The accompanying audio contains

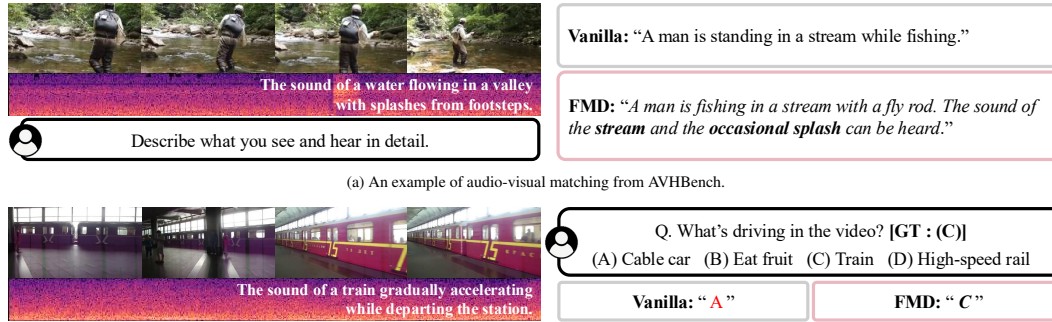

(a) An example of audio-visual matching from AVHBench.

(b) An example of audio-visual question & answering from AVQA.

Figure 3: **Qualitative results with VideoLLaMA2 on AVHBench and AVQA.** Vanilla decoding often relies on a single modality, resulting in incomplete or inconsistent outputs, whereas FMD effectively integrates both audio and visual information to produce more accurate and coherent results.

the sound of a train gradually accelerating as it departs from the station. The vanilla model, relying mainly on the ambiguous visual information, incorrectly answers (A) Cable car. In contrast, FMD correctly identifies the scene as a (C) Train, as it integrates both visual and auditory cues. This illustrates that FMD effectively leverages audio information to resolve visual ambiguity, resulting in more accurate and context-aware answers. More qualitative results, highlighting the ability of FMD to capture both audio and visual information, can be provided in the Supp. B.

### 4.4 FURTHER ANALYSIS

**Layer selection for merge point.** To determine the optimal layer for merging hidden states, we first measure the similarity across all hidden states of VideoLLaMA2, following the approach proposed in (Sun et al., 2025b). Their study on LLMs shows that layers can typically be organized into 4–5 clusters: 1–2 clusters in the early stage, a large cluster in the middle, and 1–2 clusters in the later stage. In the early layers, modalities align in feature space, with intra-modal encoding strengthened and inter-modal interaction suppressed. In the middle layers, deeper modality fusion occurs, while in the later layers, it prepares task-specific outputs. As shown in Figure 4, we observe a similar clustering pattern in VideoLLaMA2. Based on this observation, we select the fork layer $L_{fork}$ from the early stage, where feature alignment and intra-modal encoding occur (Wei et al., 2024; Yu & Lee, 2025). This choice matches our decoding strategy, aiming to strengthen the independent intra-modal representations of vision and audio. We provide analogous analysis for video-SALMONN in Supp. A.4.

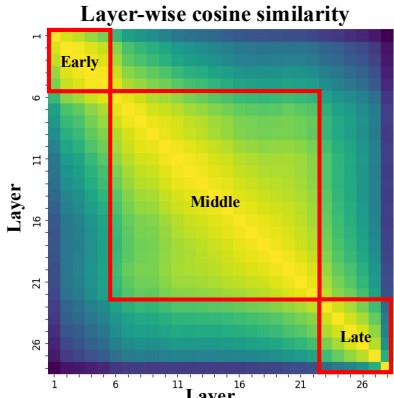

Figure 4: **Layer-wise hidden state similarity in VideoLLaMA2.** $L_{fork}$ is chosen from the early stage.

**Analysis of attention and model performance across different merge layers.** To validate the suitability of the chosen fork layer $L_{fork}$ for subsequent merging, we further analyze the attention weight distribution across layers, as illustrated in Figure 5. We find that deeper merge positions lead to reduced attention to visual tokens and increased attention to audio tokens. However, forcing equal attention to video and audio can degrade performance because it deviates from the characteristics of the pretrained model. To examine this, we analyze task performance across different *merge* layers in Figure 6, using 200 AVHBench samples for each task. We also provide results with 500 and 1,000 samples in Supp. A.5, which show similar trends. We observe that as the *merge* layer becomes deeper, the performance on A→V and AV matching tasks decreases, while performance on the V→A task improves. This implies that overly low visual dependence can hinder the interpretation of visual information. Overall, performing the *fork* operation in the early layers achieves more balanced performance while largely preserving the original model characteristics. We propose a pipeline where forking occurs early and merging starts in the middle layers, promoting unimodal feature enhancement first, followed by cross-modal interaction and reasoning in later stages.

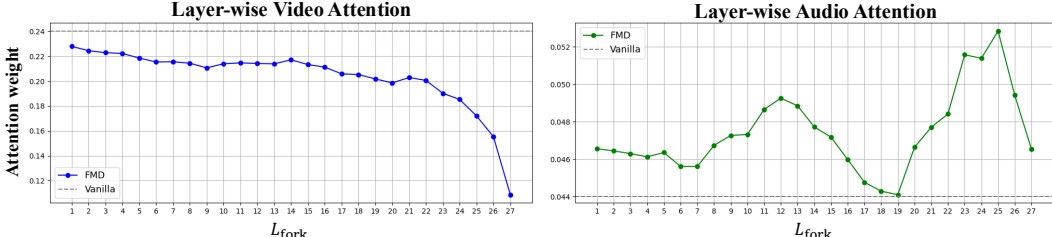

Figure 5: **Layer-wise attention weight comparison on VideoLLaMA2 using 600 samples from the AVHBench dataset.** We analyze the attention weights from the final token in the last decoder layer, focusing on the distribution across video and audio segments. Deeper merging within the network results in reduced attention to visual tokens and heightened attention to audio tokens.

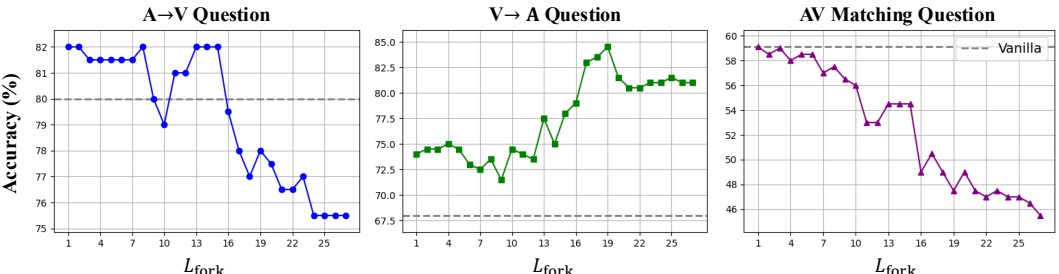

Figure 6: **Layer-wise ablation results on VideoLLaMA2 using 200 samples from the AVHBench dataset for each task.** To verify the suitablity of the selected fork layer $L_{fork}$, we evaluate performance across three tasks, each focused on a certain modality: A→V for video-targeted understanding, V→A for audio-targeted understanding, and AV matching for joint audio-visual understanding.

**Ablation on audio-visual fusion strategy.** To demonstrate the effectiveness of our attention-guided fusion, we compare it with three alternative audio-visual fusion strategies on the AVHBench dataset using the VideoLLaMA2 model in Table 3. In Eq. 3, the aggregation of masked and unmasked features is controlled by the fusion weight $\alpha$. **Exclusion** corresponds to setting $\alpha = 1$, where the masked modality is entirely excluded. This leads to a performance drop compared to vanilla decoding. We attribute this to the causal nature of autoregressive models: fully ignoring one modality can disrupt the

Table 3: **Ablation of fusion strategies in Eq. 3 with VideoLLaMA2.** Attention-guided fusion (Fixed) balances masked and unmasked inputs, improving performance.

| Methods | AVHBench | | |
|---|---|---|---|
| | A→V | V→A | AV Matching |
| Vanilla | 80.02 | 77.03 | 57.75 |
| Exclusion | 79.38 | 76.79 | 54.83 |
| Average | 75.11 | 55.31 | 57.16 |
| Attention (Adaptive) | 79.49 | 72.78 | **59.19** |
| **Attntion (Fixed)** | **80.45** | **77.52** | 59.01 |

flow of information from previous tokens to the next prediction. **Average** fusion, where $\alpha = 0.5$, also results in degraded performance, likely because it gives equal weight to informative signals and noisy features. Using an adaptive $\alpha$ for each input, referred to as **Attention (Adaptive)**, also causes performance drops on A→V and V→A tasks compared to vanilla decoding. Moreover, since obtaining the adaptive $\alpha$ requires an additional full forward pass as discussed in Section 3.1, the inference speed becomes $2.89\times$ slower than vanilla decoding. This indicates that the model is not robust to continuously varying $\alpha$ values and, and given the added inference cost, the approach is also inefficient. By contrast, a fixed $\alpha$ computed from only 100 AVBench samples not only enables efficient inference without requiring full forward pass to calculate last-layer attention, but also generalizes well to other datasets, as proven in Table 1. These results highlight the effectiveness of the attention-guided fusion strategy. Further analysis on the robustness of $\alpha$ estimation with respect to the number of samples is provided in Supp. A.6, showing that even as few as 10 samples are sufficient to obtain stable values.

**Decoding speed comparison.** To validate the efficiency of FMD, we compare its inference speed against three representative test-time decoding methods, SID (Huo et al., 2025), VCD (Leng et al., 2024b) and DoLA (Chuang et al., 2024). We apply each decoding method to VideoLLaMA2 and measure the time required to generate a single token, reporting both the absolute latency in seconds

per token and the relative speed normalized to vanilla decoding in Table 4. The results are obtained on 100 examples from the AVHBench dataset.

SID and VCD exhibit relatively slow decoding speed. Since they require two full forward passes, one on the original input and one on the modality-corrupted input to contrast the resulting logits, the computational cost nearly doubles. DoLA achieves faster inference than SID and VCD by leveraging intermediate layer outputs to refine the model's predictions, thereby mitigating the inefficiency of repeated full forward passes. Our proposed FMD achieves the fastest inference speed among the three methods. This efficiency stems from the fact that only the fork layers require dual forward passes, making FMD not only computationally efficient but also more effective than prior decoding methods originally developed for LLMs or VLMs, as evidenced by its superior performance reported in Table 2. Additional experiments analyzing how inference speed varies with deeper $L_{fork}$ settings are provided in Supp. A.7.

Table 4: **Decoding speed comparison.** FMD achieves faster inference since the process after $L_{\text{fork}}$ matches the original model.

| Decoding | Latency↓ (sec/token) | Relative↓ |
|---|---|---|
| Vanilla | 0.34 | 1 |
| SID | 0.71 | 2.09 |
| VCD | 0.69 | 2.03 |
| DoLa | 0.54 | 1.59 |
| **FMD (Ours)** | **0.43** | **1.26** |

## 5 RELATED WORKS

**Audio-visual large language models.** Building upon the success of LLMs, there has been a surge of interest in extending their capabilities to incorporate audio and visual modalities, with text serving as the central modality. ChatBridge (Zhao et al., 2023b) presents a text-centric modality bridging framework trained on limited paired data, whereas models such as PandaGPT (Su et al., 2023), ImageBind-LLM (Han et al., 2023), and OneLLM (Han et al., 2024) leverage unified encoders to accommodate various modalities. Other approaches (Chen et al., 2023b; Lu et al., 2024; Lyu et al., 2023; Panagopoulou et al., 2023; Zhan et al., 2024; Zhang et al., 2023) employ modality-specific encoders to better capture distinct feature spaces. To enhance spatial-temporal modeling across modalities, CAT (Ye et al., 2024) introduces a clue aggregator for cross-modal reasoning, VideoLLaMA2 (Cheng et al., 2024) utilizes a spatial-temporal convolutional connector for video synchronization, and video-SALMONN (Sun et al., 2024) proposes a multi-resolution causal Q-Former for audio-visual fusion. Meerkat (Chowdhury et al., 2024) further refines multimodal interactions by aligning audio and visual signals at multiple levels through interaction modules prior to decoding. Recently, Qwen2.5-Omni (Xu et al., 2025) is introduced as a model that perceives audio-visual inputs, generates text and natural speech, and achieves superior performance in audio-visual tasks.

**Inference-time reasoning enhancement with LLMs.** Recent efforts have explored inference-time strategies to enhance the reasoning capabilities of LLMs without additional training. Chain-of-Thought (CoT) guides LLMs to produce intermediate reasoning steps, and has been extended to VLMs through structured textual representations (Himakunthala et al., 2023; Ni et al., 2024; Zhu et al., 2023a) or modular reasoning pipelines (Sun et al., 2025a; Xu et al., 2024; Yang et al., 2023). These approaches improve interpretability and robustness through a more explicit reasoning process. In parallel, Contrastive Decoding (CD) improves inference-time decoding by comparing token-level logits between a weaker and a stronger model (Li et al., 2023b). DoLA (Chuang et al., 2024) develops this idea by contrasting early and late layer outputs within a single model to refine predictions. Recently, VCD (Leng et al., 2024b) extends CD to VLMs by injecting Gaussian noise into image inputs and contrasting the resulting biased predictions with the original outputs. Other CD-based approaches (Kim et al., 2024; Wang et al., 2024) enhance decoding robustness by utilizing self-descriptions or distorting instructions. SID (Huo et al., 2025) further advances this line of work by preserving the least informative visual tokens and contrasting their influence on predictions. However, most of these inference-time reasoning methods have been developed for VLMs, while AV-LLMs remain relatively underexplored. This gap underscores the need for inference strategies specifically designed to address the unique challenges of audio-visual inputs.

# 6 CONCLUSION

We analyze modality bias in current AV-LLMs, where jointly processing audio and visual inputs can hinder balanced reasoning. To address this, we propose Fork-Merge Decoding (FMD), a simple, training-free, efficient, and model-agnostic inference strategy that separates modality-specific understanding in the early decoder layers (the *fork* phase) and merges their representations in later layers (the *merge* phase). FMD consistently improves performance on tasks requiring integrated multimodal understanding, as demonstrated across three audio-visual benchmarks using three representative AV-LLMs: VideoLLaMA2, video-SALMONN, and Qwen2.5-Omni. Our approach is broadly applicable to AV-LLMs, offering a plug-and-play solution that enables deeper unimodal and multimodal understanding during inference. We hope this work inspires further research addressing the unique challenges of AV-LLMs in complex, multi-sensory settings.

**Limitations.** While our method demonstrates strong generalization across datasets without requiring ground-truth labels—relying only on 100 random samples to determine $\alpha$—we note that the optimal value of $\alpha$ can vary depending on model-specific characteristics (e.g., $\alpha = 0.8$ for VideoLLaMA2 and $\alpha = 0.9$ for Qwen2.5-Omni). This suggests that calibrating $\alpha$ is a lightweight yet necessary step when applying FMD to different architectures, and future work could explore automated strategies to further reduce this effort. In addition, although we identify fork layers that generally perform well across tasks and provide extensive analysis, the optimal choice of fork layer may still vary by task, as illustrated in Figure 6. Investigating adaptive or task-specific fork layer selection strategies therefore represents an interesting direction for future research.

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

# Supplementary Material: Fork-Merge Decoding

This supplementary material extends the main paper by including the following sections. To facilitate reproducibility, we provide the source code accompanied by a README file.

## A  FURTHER ANALYSIS

### A.1  ANALYSIS OF ATTENTION WEIGHTS BEYOND THE MERGE POINT

As shown in Figure 1 of the main paper, the final layer of VideoL-LaMA2 (Cheng et al., 2024) assigns higher attention weights to video inputs compared to audio inputs. To examine attention trends across the entire model, we visualize the average attention weights after the merge point at $L_{\text{fork}}$ in Figure A.1. Specifically, we focus on the last token in the sequence and aggregate attention across layers and heads over video and audio segments to estimate each modality's contribution to the prediction. The proposed FMD consistently produces a more balanced attention distribution between modalities.

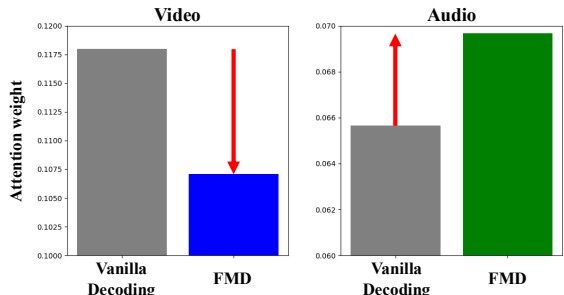

Figure A.1: **Attention weight analysis in VideoLLaMA2 on the AVHBench dataset.** We analyze 100 samples and examine the attention weights in decoder layers after $L_{\text{fork}}$, focusing on the final token. FMD narrows the gap between audio and video attention weights, encouraging more balanced contributions from both modalities.

### A.2  EVALUATION ON VIDEO-SALMONN 2+

To further validate the robustness and generalizability of our FMD approach, we evaluate its efficacy on the recent state-of-the-art AV-LLM, video-SALMONN 2+ (Tang et al., 2025). As presented in Table A.1, we assess performance on the AVHBench and World-Sense benchmarks. Note that our reported AVHBench score corresponds to the overall accuracy computed over the entire dataset, which comprises three subtasks: audio-driven video hallucination (A→V), video-driven audio hallucination (V→A), and audio-visual matching. The results demonstrate that FMD provides a consistent performance boost across both datasets. These results underscore the strong generalizability of our method.

Table A.1: **Performance of video-SALMONN 2+ on AVHBench and WorldSense datasets.** Comparison of the baseline (Vanilla) decoding strategy and our proposed approach.

| Model | AVHBench | | WorldSense | |
|---|---|---|---|---|
| | Vanilla | **FMD** | Vanilla | **FMD** |
| video-SALMONN 2+ | 65.07 | **65.70** | 46.43 | **46.81** |

### A.3  COMPARISON BETWEEN GAUSSIAN NOISE ADDITION AND ZERO-OUT MASKING

We analyze two input perturbation strategies that aim to suppress modality-specific information by replacing the original input: Gaussian noise injection, as proposed in VCD (Leng et al., 2024b), and zero-out masking, as employed in our method. Specifically, we apply each perturbation method to the video and audio inputs, respectively, and compute the cosine similarity between the final-layer hidden states of the perturbed inputs and those of the original inputs.

As shown in Figure A.2, injecting Gaussian noise into the video input yields hidden states that remain highly similar to those of the original input, indicating that this approach fails to effectively isolate the visual signal. In contrast, when applied to audio inputs, Gaussian noise successfully disrupts the signal, leading to substantial differences in the hidden representations. This observation supports the performance improvement observed in the V→A direction under the FMD with noise setting, as reported in Table 2.

In comparison, zero-out masking completely suppresses both video and audio signals, resulting in hidden states that are clearly separated from those of the original input. This demonstrates that zero-out masking more effectively blocks modality-specific information from being encoded and consistently outperforms vanilla decoding across all tasks, as shown in Table 2.

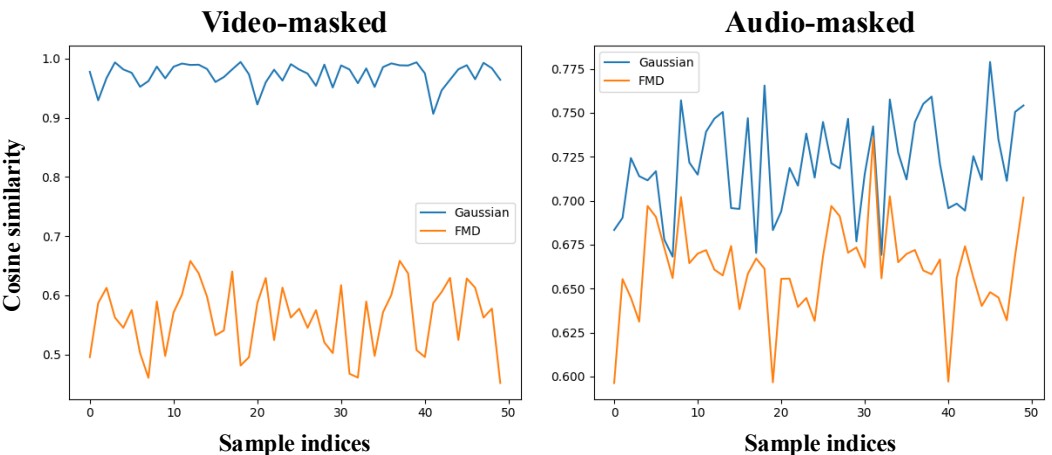

Figure A.2: **Cosine similarity comparison on VideoLLaMA2 using 100 samples from the AVH-Bench dataset.** We compute the cosine similarity between the final-layer hidden states of the intact input and those of audio- or video-perturbed inputs. The results indicate that video signals are not effectively isolated by additive Gaussian noise, whereas zero-masking reliably suppresses both modalities.

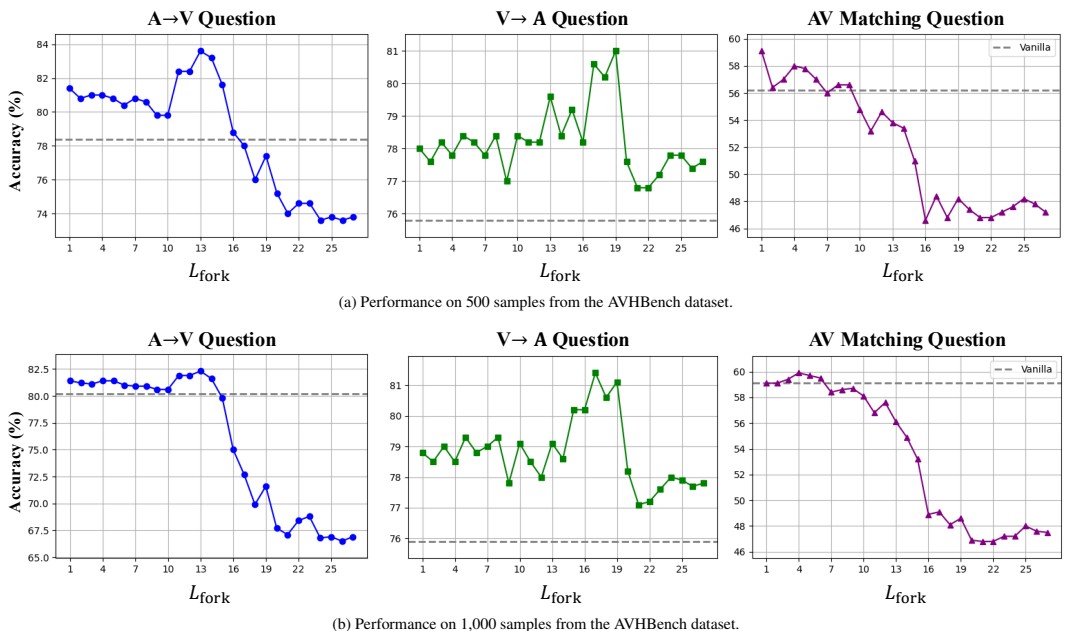

Figure A.3: **Layer-wise accuracies of VideoLLaMA2 on each task in the AVHBench dataset.** The consistent trends observed across 500 and 1,000 samples further validate the 200-sample analysis presented in Figure 6.

### A.4 LAYER-WISE HIDDEN-STATE SIMILARITY ANALYSIS FOR CHANNEL-WISE FUSION

To clarify the criterion for selecting $L_{\text{fork}}$, we apply the same cosine-similarity analysis used in Figure 4 of the main paper to the channel-wise fusion model, video-SALMONN (Sun et al., 2024). As shown in Figure A.4, video-SALMONN exhibits the same progression observed in the main paper: the early layers form one to two clusters, the middle layers collapse into a single cluster as multimodal features are integrated, and the later layers again split into one to two clusters. This consistent early–middle–late pattern directly supports our choice of $L_{\text{fork}}$, since placing it before the middle-layer fusion allows modality-specific representations to be refined prior to being merged. Based on this analysis, we select $L_{\text{fork}} = 6$ for video-SALMONN.

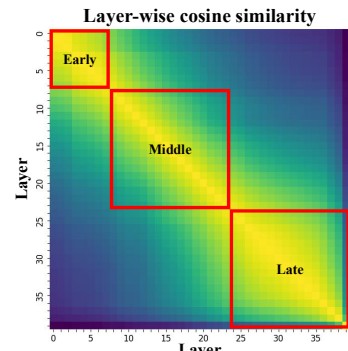

Figure A.4: **Layer-wise hidden state similarity in video-SALMONN.** $L_{\text{fork}}$ is chosen from the early stage.

### A.5 LAYER-WISE PERFORMANCE ACROSS DATASET SIZES

In the main paper (see Section 4.4 and Figure 6), we analyze task performance across $L_{\text{fork}}$ using 200 samples from the AVHBench dataset. To examine whether this trend holds with more data, we repeat the analysis with 500 and 1,000 samples per task (see Figure A.3a and Figure A.3b). We observe consistent trends across all sample sizes, demonstrating that $L_{\text{fork}}$ can be reliably validated even with a small number of samples.

### A.6 SENSITIVITY OF $\alpha$ TO SAMPLE SIZE

To examine how many samples are required to obtain a reliable estimate of the attention weight $\alpha$ used for attention-guided fusion (Section 3.1), we evaluate its robustness under realistic deployment scenarios with the VideoLLaMA2 model. While the main paper computes $\alpha$ using 100 samples, in practice the available data may be far smaller. To simulate this setting, we first combine examples from the three datasets (AVHBench, AVUT, and WorldSense) into a single pool. From this pool, we randomly sample 10, 50, or 100 examples, repeating each sampling 100 times to assess the stability of $\alpha$. As reported in Table A.2, the estimated $\alpha$ remains highly stable even when only 10 samples are used, and the variance decreases further with larger subsets. This indicates that our procedure for estimating $\alpha$ is robust and data-efficient, enabling reliable parameter selection even with very limited data in real-world deployment.

Table A.2: **Stability of $\alpha$ values based on sample size.**

| Number of Samples | 10 | 50 | 100 |
|---|---|---|---|
| Mean ($\mu$) | 0.829 | 0.829 | 0.829 |
| Std ($\sigma$) | 0.024 | 0.010 | 0.007 |

### A.7 ANALYSIS OF INFERENCE SPEED ACROSS $L_{fork}$

To better understand the relationship between inference speed and the fork layer $L_{fork}$, we evaluate FMD on VideoLLaMA2 using 100 randomly selected AVHBench samples across different fork layer settings.

As shown in Table A.3, deeper fork layers require more decoder layers to be forwarded, leading to longer inference time. At the same time, accuracy drops significantly because later forking disrupts the pretrained model attribute as illustrated in Figure 5 and Figure 6. Based on this, we set $L_{fork} = 4$ in VideoLLaMA2 (one out of every seven layers), which yields consistent performance gains while increasing inference time by only 26%.

## B MORE QUALITATIVE RESULTS

In addition to the qualitative analysis presented in Section 4.3, we include further examples that illustrate both successful outcomes and failure cases, providing a more comprehensive understanding of the behavior of the model.

Table A.3: **Analysis between inference speed and fork layer $L_{fork}$ in VideoLLaMA2 on AVH-Bench.** Deeper fork layers require more decoder computation, increasing latency while significantly reducing accuracy as illustrated in Figure A.3. Setting $L_{fork} = 4$ (one-seventh of the 28 layers) achieves consistent performance gains with only 26% increase in inference time.

| $L_{fork}$ | 0 (Vanilla) | 1 | **4** | 5 | 10 | 15 | 20 | 25 |
|---|---|---|---|---|---|---|---|---|
| Latency (sec/token) | 0.34 | 0.40 | **0.43** | 0.45 | 0.51 | 0.56 | 0.62 | 0.68 |
| Relative | 1 | 1.18 | **1.26** | 1.32 | 1.5 | 1.65 | 1.82 | 2 |

### B.1 POSITIVE CASES

In addition to Figure 3, we further analyze various cases, including audio-driven video hallucinations, video-driven audio hallucinations, and audio-visual matching, as illustrated in Figure A.5, Figure A.6, and Figure A.7. We also examine complex audio-visual description scenarios in Figure A.8. In these cases, our proposed FMD effectively leverages both audio and visual modalities to accurately interpret the contexts. Notably, even when the audio and video are artificially constructed, FMD is able to detect inconsistencies and generate correct responses. Moreover, as shown in Figure A.8, FMD outperforms vanilla decoding by producing more precise and detailed descriptions of both modalities.

### B.2 NEGATIVE CASES

Although our FMD significantly enhances multimodal understanding without requiring additional training, it occasionally produces inaccurate phrases alongside otherwise detailed and accurate descriptions. For example, in the top case of Figure A.9, FMD successfully captures both visual and audio content, whereas vanilla decoding describes only the visual scene. However, the phrase "while she does it" is temporally incorrect, since the speaker talks before blow-drying. In the bottom example, the mention of "cup" is inaccurate, as no cup is present. Although such cases reflect occasional misunderstandings, FMD generally produces richer and more informative responses by modeling cross-modal relationships (e.g., capturing "talking"). Moreover, at a broader scale, it tends to reduce hallucinations across datasets, thereby improving overall performance, as demonstrated in Table 1. Further investigation into minimizing fine-grained hallucinations remains an important direction for future work.

## C COMPUTATIONAL RESOURCE

All experiments are conducted on a system equipped with an AMD EPYC 7513 32-Core CPU and a single NVIDIA RTX A6000 GPU. To ensure fair measurement of inference speed across all experiments, we terminate all non-experimental processes during inference time.

## D THE USE OF LLMS

We use LLMs to refine words and sentences for a formal academic writing style and to identify relevant related work. For evaluation, we employ LLMs to assess long-form responses, such as AV captioning from AVHBench, following the protocol of VideoLLaMA2.

## E SOCIAL IMPACT

The rapid advancement of LLMs has significantly influenced various sectors, including technology, culture, and education, by making information more accessible and enabling efficient communication. In parallel, MLLMs have progressed through the integration of visual modalities into LLM decoders. Recently, AV-LLMs have emerged, extending these capabilities to both visual and auditory content.

By directly understanding and reasoning over audio-visual content, AV-LLMs offer practical benefits in everyday settings where information is naturally multimodal, such as videos, lectures, conversations, and real-world environments. This opens new possibilities for applications like multimodal search, video question answering, and assistive technologies that go beyond text-based interfaces.

Much like how LLMs have made text-based knowledge more accessible, AV-LLMs can help users navigate and interact with rich multimedia content more effectively.

Despite these advantages, the inference-time behavior of AV-LLMs remains underexplored. Our proposed Fork-Merge Decoding (FMD) provides a training-free framework to analyze and isolate modality-specific reasoning, offering insights into how AV-LLMs process complex audio-visual information and guiding future improvements in model design and interpretability.

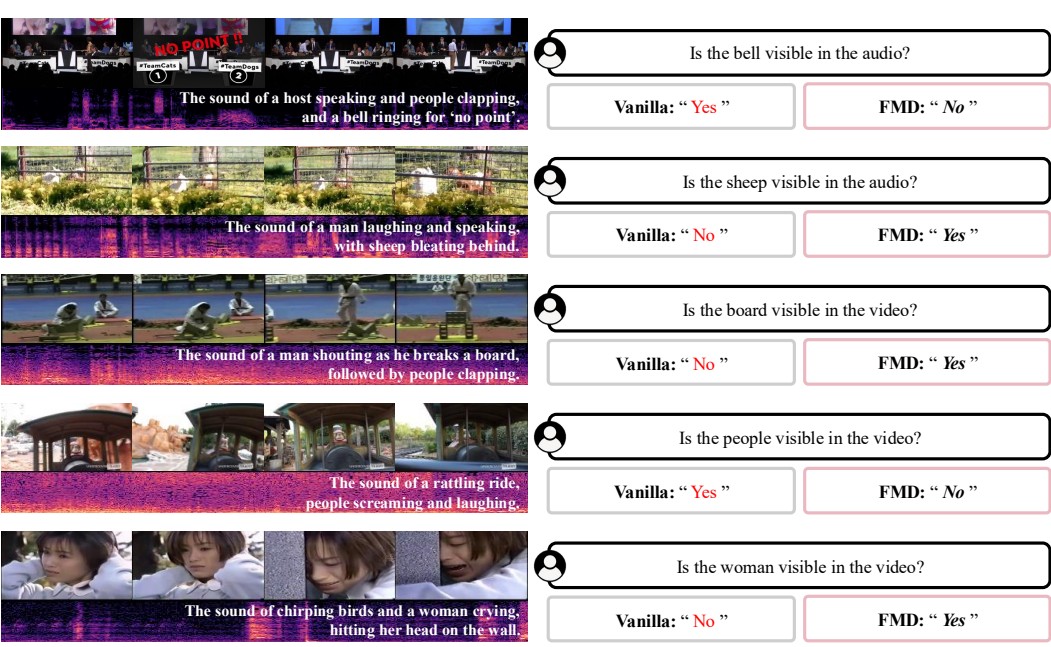

Figure A.5: **Qualitative results for audio-driven video hallucination tasks using VideoLLaMA2.** Compared to vanilla decoding, FMD generates more accurate responses by effectively leveraging both audio and visual modalities.

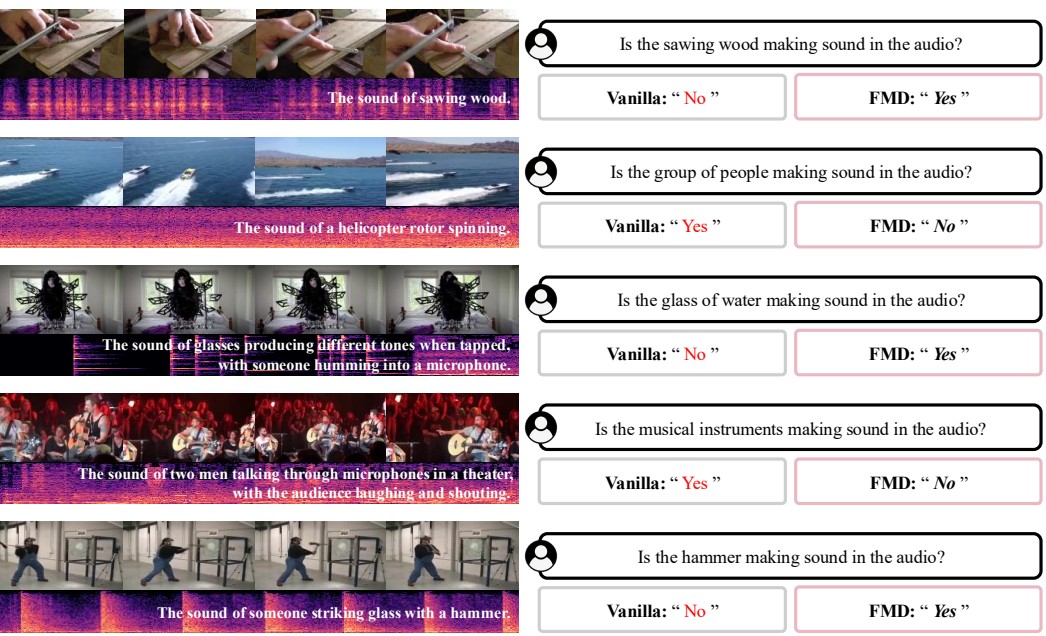

Figure A.6: **Qualitative results for video-driven audio hallucination tasks using VideoLLaMA2.** FMD produces more accurate responses in most cases, including instances where visual signals lead to confusion, as shown in the third example.

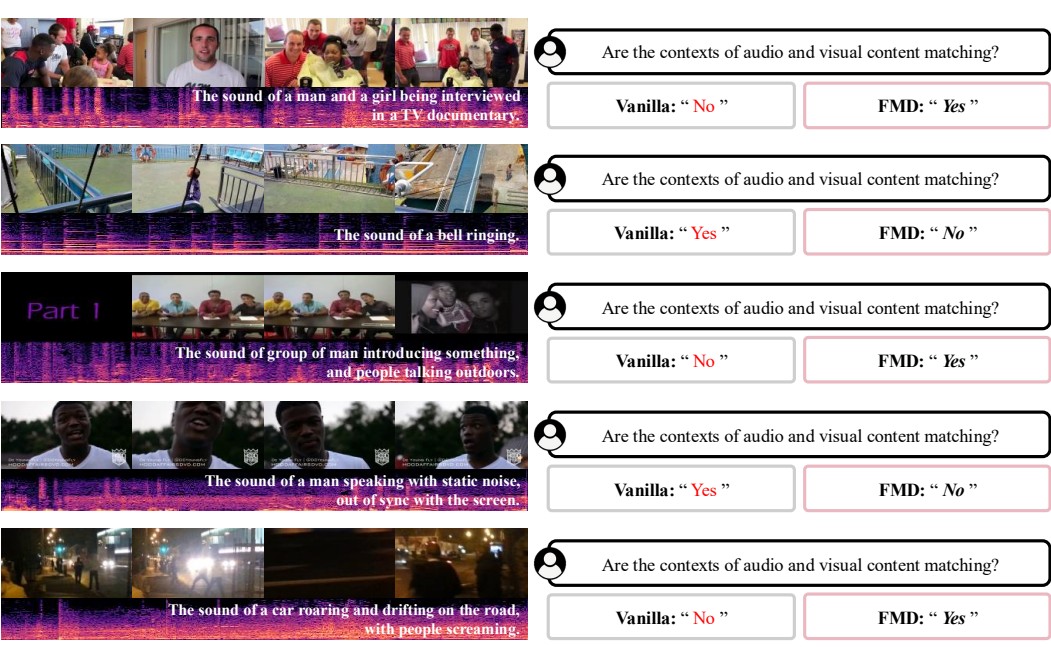

Figure A.7: **Qualitative results for audio-visual matching tasks using VideoLLaMA2.** FMD generates correct responses even when the synthesized audio and video inputs are semantically unrelated.

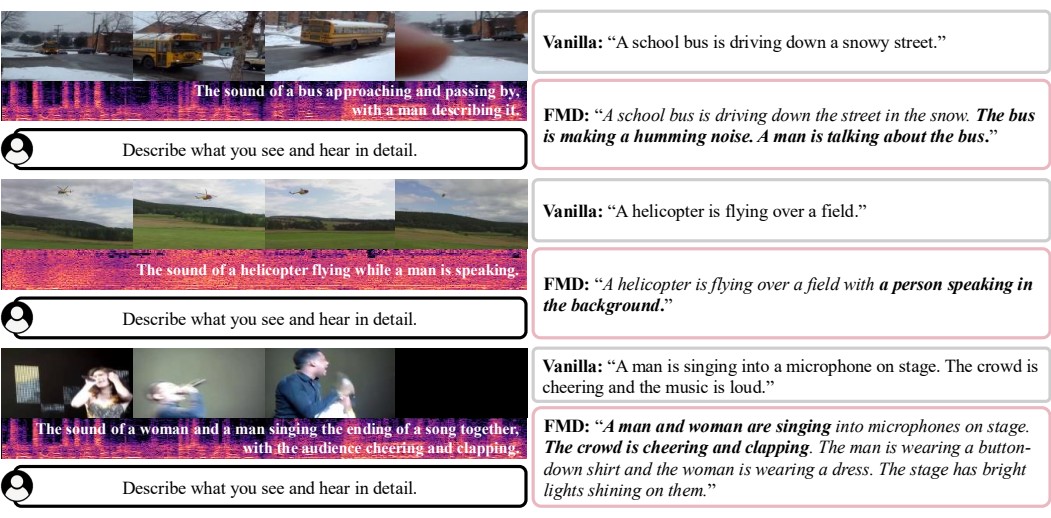

Figure A.8: **Qualitative results for audio-visual description tasks using VideoLLaMA2.** FMD effectively describes both audio and visual content, capturing fine-grained details (bold)—such as "A man is talking about the bus" in the top example.

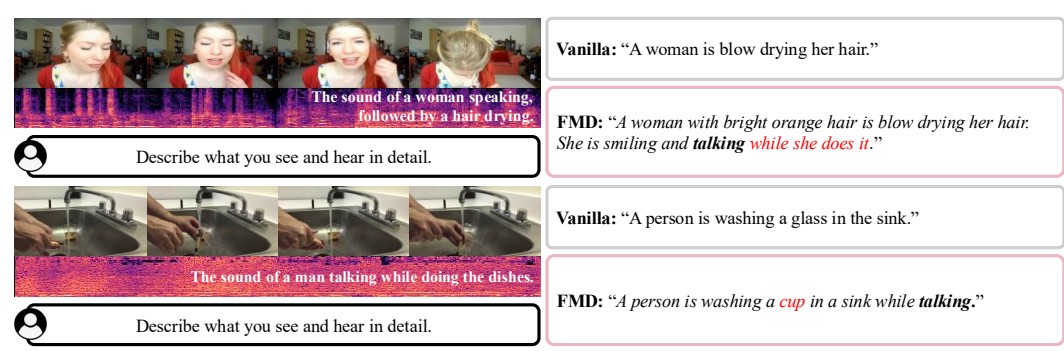

Figure A.9: **Failure case analysis on audio-visual description tasks with VideoLLaMA2 (Cheng et al., 2024).** FMD may produce occasional errors (red), but captures both audio and visual content more effectively and with finer-grained details (bold) than vanilla decoding.

