# OpenReview forum: "Fork-Merge Decoding: Enhancing Multimodal Understanding in Audio-Visual Large Language Models"
_ICLR.cc/2026/Conference — Submitted to ICLR 2026_

### Official Review · Reviewer_4GtS · 2025-10-23

**Soundness:** 3
**Presentation:** 3
**Contribution:** 3
**Rating:** 6
**Confidence:** 4

**Summary:**

This paper proposes the fork-merge decoding method to improve audio-visual LLM performance in a training free way. The process first masks audio or visual inputs separately in earlier TFM layers, and then merge them in later layers so that the model is not overlooking any modality when answering questions.

**Strengths:**

1. The method is training free and trying to solve a critical problem in audio-visual LLMs

2. The proposed method achieved superior performance on a range of benchmarks over 3 different audio-visual LLMs in the experiments.

3. The paper is very well-written.

**Weaknesses:**

I mainly have the following two concerns regarding the experimental setup, and require the authors to provide more information:

(1). Including more difficult benchmarks. I would suggest at least adding results on Video-MME [1] and/or AVUT [2] to demonstrate the consistency of improvements and the balance between audio and visual modalities. No need to do for all 3 models but at least 1 model is needed.

(2). More recent models, e.g. video-SALMONN 2+ [3] which achieved the best performance on Video-MME would be very helpful to show that the state-of-the-art models still benefit from this method.

[1] Fu et al. "Video-MME: The First-Ever Comprehensive Evaluation Benchmark of Multi-modal LLMs in Video Analysis", https://arxiv.org/abs/2405.21075

[2] Yang et al. "Audio-centric Video Understanding Benchmark without Text Shortcut", https://arxiv.org/abs/2503.19951

[3] Tang et al. "video-SALMONN 2: Caption-Enhanced Audio-Visual Large Language Models", https://arxiv.org/abs/2506.15220

**Questions:**

1. Have the authors tried soft masking?

2. How much additional computational cost does the merge decoding process introduce?

3. Any insights into how the model improves on hallucination induced by not having balanced modality attention? My experience is that audio-visual LLMs may make up things on a modality that they fail to attend to, so any insights?

---

> ### Author Response · Authors · 2025-11-21
> **Official Comment by Authors**
>
> Thanks to the reviewer’s insightful comments, we were able to add detailed information that enhances reader understanding. For your convenience, all revisions and additions have been clearly highlighted in purple in the updated PDF. Based on the comments, we have:
>
> > **(1)** evaluated a more recent model and additional benchmarks to further validate the effectiveness of FMD,
> > **(2)** provided results for the soft-masking variant,
> > **(3)** clarified the additional computational cost introduced by the two fork processes,
> > **(4)** shared thoughts on how FMD helps mitigate hallucination.
>
>
> Our detailed responses are provided below.
>
> ---
>
> **[W1, W2] Requests for recent state-of-the-art model and more difficult benchmarks.**
>
> To demonstrate robustness, we incorporated FMD into the recent video-SALMONN 2+ **[1]**, as suggested. Evaluation on AVHBench and WorldSense **[2]** confirms that FMD consistently improves performance, even when applied to such a strong state-of-the-art baseline (see Table below).
>
> | **Model**         | **AVHBench (Vanilla / FMD)** | **WorldSense (Vanilla / FMD)** |
> |-------------------|:----------------------------:|:-------------------------------:|
> | video-SALMONN 2+  | 65.07 / **65.70**            | 46.43 / **46.81**               |
>
> To further validate the generalizability of our approach, we expanded our evaluation to include two additional benchmarks, AVUT **[3]** and WorldSense **[2]**. By applying the same baselines used in the main paper (VideoLLaMA2 and video-SALMONN), we observed consistent performance gains with FMD (see Table below), thereby confirming its robustness across diverse datasets.
>
> | **Model**       | **AVUT (Vanilla / FMD)** | **WorldSense (Vanilla / FMD)** |
> |-----------------|:-------------------------:|:-------------------------------:|
> | VideoLLaMA2     | 44.44 / **45.13**         | 29.41 / **29.91**               |
> | video-SALMONN   | 32.36 / **32.85**         | 56.20 / **57.83**               |
>
>
> > **[Reference]**
> > **[1]** *Video-SALMONN 2: Caption-Enhanced Audio-Visual Large Language Models, arXiv 2025.*
> > **[2]** *WorldSense: Evaluating Real-world Omnimodal Understanding for Multimodal LLMs, arXiv 2025.*
> > **[3]** *Audio-centric Video Understanding Benchmark without Text Shortcut, EMNLP 2025.*
>
> ---
>
> **[Q1] Have the authors tried soft masking?**
>
> We have explored soft masking. In the main paper (**Table 2**), we compare the performance of Gaussian-noise–based soft masking and zero-out masking, demonstrating the effects of each approach on model performance. Furthermore, as shown in **Figure A.2**, the features obtained after adding Gaussian noise to the original video or audio tokens remain highly similar to the original features, indicating that the model can still access a substantial amount of the original modality information. As the soft masking method does not sufficiently suppress the masked modality, it fails to enforce the intended separation and ultimately results in degraded performance compared with our zero-out strategy.
>
> ---
>
> **[Q2] How much additional computational cost does the merge decoding process introduce?**
>
> The additional computational cost is limited because we set only the first 1/7 of LLM layers as fork layers. This results in an extra computation 1/7 of the original LLM forward pass.
>
> ---

---

> ### Author Response · Authors · 2025-11-21
>
> ---
>
> **[Q3] What insights explain how the model mitigates hallucinations that occur due to unbalanced modality attention?**
>
> FMD mitigates hallucination not by explicitly correcting errors, but by ensuring the model constructs deeper unimodal reasoning paths before multimodal fusion.
> To verify that the modality-masked hidden states (unimodal paths) still contain meaningful representations, we conduct an additional captioning experiment. Specifically, VideoLLaMA2 is queried with audio-masked and video-masked inputs from AVHBench, conditioned on the prompt "Describe the given video and audio." (See the table below).
>
> The results clearly show that the model consistently focused on the available modality: audio-masked inputs resulted in video-only descriptions, and vice versa. Building on this observation, we leverage the early layers (fork stage) to cultivate deeper unimodal understanding before merging the features for multimodal reasoning in later layers. This strategy effectively mitigates hallucination, as evidenced by the performance gains on AVHBench.
>
> **Prompt: "Describe what you see and hear in a single sentence."**
>
> | Audio-masked                                                  | Video-masked                                   |
> |---------------------------------------------------------------|-------------------------------------------------|
> | "A dog is playing with a tire on a rug."                      | "A dog barks and growls."                      |
> | "A woman in a life jacket is standing on a rock in the water."| "A woman is speaking and water is splashing."  |
> | "A car engine with a blue wire connected to it."              | "An engine is knocking."                       |
> | "A baby sits on a table and looks at her mother."             | "A baby is babbling and a woman is laughing."  |
> | "A video game displays a large ship with a large alien on it."| "A video game sound plays."                    |
> | "Bee hive in the woods."                                      | "Bee is buzzing."                              |
> | "Two frogs are sitting on a tree branch."                     | "Frogs are croaking."                          |
> | "A man is riding a yellow quad bike through the woods."       | "A motorcycle is driving."                     |
> | "A man is walking through corn fields with a gun while birds fly in the sky." | "A cap gun is fired and a bird chirps." |
> | "A toilet bowl with water in it."                             | "Toilet flushing with water."                  |
>
> ---

---

### Official Review · Reviewer_wsv4 · 2025-10-29

**Soundness:** 3
**Presentation:** 3
**Contribution:** 2
**Rating:** 4
**Confidence:** 4

**Summary:**

This paper introduces Fork-Merge Decoding (FMD), a training-free inference strategy designed to mitigate modality bias in Audio-Visual Large Language Models (AV-LLMs). The core problem addressed is the tendency of AV-LLMs to over-rely on one modality (e.g., video) at the expense of another (e.g., audio), leading to imbalanced understanding and errors. The authors validate FMD on three representative AV-LLMs (VideoLLaMA2, video-SALMONN, Qwen2.5-Omni) across three challenging benchmarks (AVQA, MUSIC-AVQA, AVHBench). The results consistently show that FMD improves performance over vanilla decoding, particularly in tasks requiring balanced audio-visual reasoning. However, the performance improvement offered by this method is not particularly significant, and some of its design choices lack robust experimental validation.

**Strengths:**

- FMD is convenient to deploy; as a training-free, plug-and-play method, it is adaptable to a wide range of applications.
- The comprehensive ablation studies presented in the paper demonstrate the authors' thoroughness in validating their design.
- The proposed method is simple and intuitive, and the manuscript is of high writing quality.

**Weaknesses:**

Although the paper includes extensive ablation studies, I suggest that the following aspects require further investigation:
- I would like to clarify if the model employs Nucleus Sampling for decoding. If it does, given its stochastic nature, it is crucial to conduct multiple decoding runs and report the results with error bars. This analysis is essential because the performance improvement from FMD is currently marginal, which raises concerns about the stability and statistical significance of the method.
- For the model using CHANNEL-WISE FUSION, there is a lack of detailed experimental analysis regarding the audio and video weights during the merging stage. The majority of the analysis in the paper is based on VideoLLaMA-2, with insufficient corresponding analysis for the CHANNEL-WISE FUSION model.
- More powerful models should also be evaluated, such as the recently released video-SALMONN 2+, Qwen3-Omni.
- More benchmarks should be included, such as Video-MME, WorldSense, AVUT.

**Questions:**

I agree with the authors that the current choice of $\alpha$ and the fork layer is suboptimal. I would be interested to know if the authors have any thoughts on potential improvements or alternative approaches. This question will not affect my overall assessment of the paper.

---

> ### Author Response · Authors · 2025-11-21
>
> Thanks to the reviewer’s constructive comments, We were able to add important details to imporve the clarity of our work. For your convenience, all revisions and additions have been clearly highlighted in purple in the updated PDF. Based on the comments, we have:
>
> > **(1)** clarified the decoding type of the AV-LLMs used in the experiments,
> > **(2)** provided further analysis on channel-wise fusion method,
> > **(3)** evaluated a more recent model and additional benchmarks to further verify the effectiveness of FMD.
>
> Our detailed responses are provided below.
>
> ---
>
> **[W1] Does the decoding process involve any stochastic behavior?**
>
> All models in our experiments use standard deterministic decoding methods, namely greedy decoding or beam search, rather than sampling based approaches such as nucleus sampling. VideoLLaMA2, Qwen2.5-Omni, and video-SALMONN 2+ **[1]** use greedy decoding, while video-SALMONN uses beam search. We will clarify this detail in the final version to avoid any misunderstanding.
>
> > **[Reference]**
> > **[1]** *Video-SALMONN 2: Caption-Enhanced Audio-Visual Large Language Models, arXiv 2025.*
>
> ---
>
> **[W2] Why does the analysis focus on token-wise fusion, and where is the analysis for channel-wise fusion?**
>
> In the channel wise model, audio and video features are entangled within the same hidden states rather than represented as separate modality specific tokens. Because there are no separate audio and video token groups, the modality specific attention analysis used for token wise models cannot be applied to this architecture.
> To address the reviewer’s concern, we additionally provide the hidden state similarity analysis for the channel wise model, following the same method already used for the token wise case. We include these new results in the revised supplementary material as **Figure A.4**. The observed trends align with those of the token-wise model and further support our fork-layer selection.
>
> ---
>
> **[W3, W4] Requests for powerful models and more challenging benchmarks**
>
> To validate the robustness of FMD, we apply it to the powerful AV-LLM, video-SALMONN 2+ **[1]**, as suggested. Experiments are conducted on both AVHBench(A→V, V→A, AV matching), and WorldSense datasets. As shown in the table below, FMD consistently improves the performance of the state-of-the-art AV-LLM on both benchmarks.
>
> | **Model**             | **AVHBench (Vanilla / FMD)** | **WorldSense (Vanilla / FMD)** |
> |-------------------|:-------------------------:|:----------------------------:|
> | video-SALMONN 2+  | 65.07 / **65.70**             | 46.43 / **46.81**               |
>
> To verify the generalizability of our method, we extend our evaluation to two additional benchmarks, AVUT **[2]** and WorldSense **[3]**, using the same models as in the main paper (VideoLLaMA2 and video-SALMONN). As shown in the table below, FMD yields consistent improvements on these datasets, underscoring its robustness.
>
> | **Model**         | **AVUT (Vanilla / FMD)** | **WorldSense (Vanilla / FMD)** |
> |---------------|:----------------------:|:----------------------------:|
> | VideoLLaMA2   | 44.44 / **45.13**          | 29.41 / **29.91**                |
> | video-SALMONN | 32.36 / **32.85**          | 56.20 / **57.83**                |
>
> > **[Reference]**
> > **[1]** *Video-SALMONN 2: Caption-Enhanced Audio-Visual Large Language Models, arXiv 2025.*
> > **[2]** *Audio-centric Video Understanding Benchmark without Text Shortcut, EMNLP 2025.*
> > **[3]** *WorldSense: Evaluating Real-world Omnimodal Understanding for Multimodal LLMs, arXiv 2025.*
>
> ---
>
> **[Q1] Regarding alternative ideas for choosing α and $L_{fork}$**
>
> We acknowledge that dynamic parameter selection remains a challenging open problem, as reported in our limitations.
> First, regarding $\alpha$, our initial attempts at adaptive estimation led to reduced performance, as shown in **Table 3** of the main paper.
> Similarly, for $L_{fork}$, motivated by the observation that the optimal depth varies across tasks, we explored an adaptive strategy by estimating prompt-based modality bias using commercial LLMs. While bias estimation was feasible, mapping this bias to an optimal layer depth proved highly challenging due to the sensitivity of performance to specific layers. We thus reserve this adaptive mechanism for future work.
>
> ---

---

### Official Review · Reviewer_xuKv · 2025-10-30

**Soundness:** 2
**Presentation:** 2
**Contribution:** 2
**Rating:** 4
**Confidence:** 3

**Summary:**

This paper addresses the modality bias problem in current audio-visual large language models (AV-LLMs) by proposing Fork-Merge Decoding (FMD)—a training-free, architecture-agnostic inference-time strategy. FMD separates early-stage unimodal reasoning (fork) from later-stage multimodal integration (merge). The method consistently improves performance across three representative AV-LLMs (VideoLLaMA2, video-SALMONN, and Qwen2.5-Omni) and three standard benchmarks, especially on tasks requiring balanced cross-modal reasoning. The authors further validate FMD’s compatibility with both token-wise and channel-wise fusion architectures and support its effectiveness through attention analysis, ablation studies, and qualitative examples.

**Strengths:**

1. Important and practical problem: Modality bias is a core challenge in modern multimodal LLMs; this work tackles a real-world issue with clear applicability.

2. Simple and efficient method: FMD requires no additional training or architectural changes—only input masking and hidden-state fusion during inference—making it a plug-and-play solution that is easy to deploy.

3. Comprehensive experiments: The evaluation covers diverse AV-LLM architectures and multiple benchmark datasets, demonstrating consistent gains and robustness.

**Weaknesses:**

1. Heuristic hyperparameter selection:
The fusion weight α is estimated as a fixed value from 100 samples on AVHBench (e.g., α = 0.8 for VideoLLaMA2). Although it generalizes well to other datasets, it lacks theoretical grounding or an adaptive mechanism. Manual calibration per model undermines true plug-and-play usability in real-world settings.
As shown in Figure 6, the optimal Lfork (fork layer depth) varies significantly across tasks (e.g., A→V prefers shallow layers, V→A prefers deeper ones), raising concerns about practical adaptability.

2. Inference speed overhead:
As shown in Table A.1, FMD increases latency by 1.26× compared to vanilla decoding. While modest, this overhead may limit deployment in latency-sensitive applications.

**Questions:**

1. On the generalizability of α:
How sensitive is performance to the choice of α? Specifically, if we use a suboptimal α (e.g., from a different model or dataset), how much does performance degrade? Could you quantify this drop?

2. On the meaningfulness of masked features:
In the fork phase, one modality (e.g., video tokens) is zeroed out. Yet the resulting hidden states still carry meaningful unimodal signals. Why is this the case? Does the language prompt alone provide enough context for the model to generate coherent intermediate representations—even when one sensory modality is completely masked?

3. On adaptive Lfork selection:
Figure 6 shows that A→V tasks benefit from shallow Lfork, while V→A tasks prefer deeper Lfork. Have you considered a dynamic Lfork selection mechanism—for example, based on the modality bias inferred from the input prompt or early attention patterns? Would such an adaptive strategy further improve performance?

4. On text modality bias and counterfactual inputs:
The paper focuses on audio-visual imbalance, but text is also a modality. Could the model answer correctly even when both audio and video are masked, relying only on the textual prompt? Moreover, when audio and video are counterfactual or inconsistent, does the model tend to output stereotypical or common-sense answers (biased by pretraining) rather than reflecting the actual (possibly rare) content in the video? This would reveal whether text itself introduces another form of bias.

---

> ### Author Response · Authors · 2025-11-21
>
> We sincerely thank the reviewer for the thorough and insightful feedback. For your convenience, all revisions and additions have been clearly highlighted in purple in the updated PDF. The detailed critique helped us clarify and strengthen our arguments. Based on the comments, we have:
>
> > **(1)** added experiments demonstrating the robustness of α across different sample sizes and explored adaptive $ L_{fork}$ selection,
> > **(2)** addressed concerns regarding inference speed overhead,
> > **(3)** evaluated the sensitivity of α,
> > **(4)** provided examples showing that modality-masked hidden states still yield meaningful representations,
> > **(5)** explained the effect of text bias and showed that FMD consistently improves performance even in audio–visual inconsistent cases.
>
> Our detailed responses are provided below.
>
> ---
>
> **[W1, Q3] Concerns regarding the calibration of α in real-world settings and the possibility of an adaptive strategy for selecting $L_{fork}$ .**
>
> **α**: We acknowledge that determining α involves a calibration step. However, we emphasize that this estimation requires minimal effort. By pooling samples from three diverse datasets (AVHBench, AVUT **[1]**, and WorldSense **[2]**) and testing small batches (10 or 50 samples) across 100 trials, we observe that the resulting α values are remarkably consistent, as shown in the table below. Given that α is rounded for practical use, small variances do not affect the final model configuration. This demonstrates that our selection process is not only robust but also independent of specific tasks, requiring minimal data to generalize.
>
> | **Number of Samples** | **10** | **50** | **100** |
> |------------------------|:------:|:------:|:-------:|
> | Mean                   | 0.829  | 0.829  | 0.829   |
> | Std                    | 0.024  | 0.010  | 0.007   |
>
> **$L_{fork}$**: First, the selection of $L_{fork}$ is not heuristic but grounded in our layer-wise hidden-state similarity analysis (Main Paper, **Figure 4**). This analysis consistently reveals that the early layers of the LLM function as a unimodal processing stage as explained in the main paper. Based on this observation, we deterministically set $L_{fork}$ to approximately one-seventh of the total depth. This fixed configuration proves robust across various models and datasets, balancing performance with efficiency; placing the fork deeper would significantly increase computation and latency.
>
> Additionally, we explored an adaptive $L_{fork}$ strategy by estimating prompt-based modality bias using commercial LLMs. While bias estimation was feasible, mapping this bias to an optimal layer depth proved highly challenging due to the sensitivity of performance to specific layers. As illustrated in **Figure 6**, performance fluctuates significantly even between adjacent layers; for instance, there exists a ~2% gap between layers 8 and 9 in A→V tasks, and a ~2.5% gap between layers 9 and 10 in V→A tasks. Given such noisy performance differences, developing a robust adaptive selection mechanism requires more in-depth investigation. We therefore leave this as a subject for future work and adhere to the robust fixed strategy.
>
> >**[Reference]**
> > **[1]** *Audio-centric Video Understanding Benchmark without Text Shortcut, EMNLP 2025.*
> > **[2]** *WorldSense: Evaluating Real-world Omnimodal Understanding for Multimodal LLMs, arXiv 2025.*
>
> ---
>
> **[W2] Concern regarding inference-time speed overhead**
>
> We acknowledge the additional latency, but note that FMD does not require any extra full forward pass, unlike methods such as Chain of Thought **[3]** or Contrastive Decoding **[4]**. The overhead comes only from splitting early layers, and is therefore modest and bounded.
>
> >**[Reference]**
> >**[3]** *Chain-of-Thought Prompting Elicits Reasoning in Large Language Models“, NeurIPS 2022.*
> >**[4]** *Contrastive Decoding: Open-ended Text Generation as Optimization”, ACL 2023.*
>
> ---
>
> **[Q1] How sensitive is performance to the choice of α?**
>
> To test parameter sensitivity, we used α=0.9 (optimized for Qwen2.5-Omni) on VideoLLaMA2. Even with this mismatch, FMD improved upon the baseline. This confirms that while a model-specific α (α=0.8) is ideal, the method is robust to small variations in hyperparameter selection.
>
> | **Method**            | **AVHBench** | **AVUT**  |
> |-------------------|:------------:|:--------:|
> | Vanilla           | 70.75    | 44.44 |
> | FMD w/ 0.9        | 70.82    | 45.02 |
> | FMD w/ 0.8 (Ours) | **71.51**    | **45.13** |
>
> ---

---

> ### Author Response · Authors · 2025-11-21
>
> ---
>
> **[Q2] Can the language prompt alone provide enough context to generate coherent intermediate representations when a modality is fully masked?**
>
> To verify that the modality-masked hidden states still contain meaningful unimodal representations, we conduct an additional captioning experiment. Specifically, VideoLLaMA2 is queried with audio-masked and video-masked inputs from AVHBench, conditioned on the prompt "Describe the given video and audio.".
>
> The results clearly show that the model focuses exclusively on the remaining modality: audio-masked inputs produce video-only descriptions, and video-masked inputs produce audio-only descriptions. This demonstrates that the text prompt provides sufficient context for the model to route its representations toward the available modality, even when the other modality is zeroed out at the input level.
> We will include these qualitative examples in the revised supplementary material. Ten representative cases are shown below.
>
> **Prompt: "Describe what you see and hear in a single sentence."**
> | **Audio-masked**                                                   | **Video-masked**                                  |
> |--------------------------------------------------------------------|----------------------------------------------------|
> | "A dog is playing with a tire on a rug."                           | "A dog barks and growls."                         |
> | "A woman in a life jacket is standing on a rock in the water."     | "A woman is speaking and water is splashing."     |
> | "A car engine with a blue wire connected to it."                   | "An engine is knocking."                          |
> | "A baby sits on a table and looks at her mother."                  | "A baby is babbling and a woman is laughing."     |
> | "A video game displays a large ship with a large alien on it."     | "A video game sound plays."                       |
> | "Bee hive in the woods."                                           | "Bee is buzzing."                                 |
> | "Two frogs are sitting on a tree branch."                          | "Frogs are croaking."                             |
> | "A man is riding a yellow quad bike through the woods."            | "A motorcycle is driving."                        |
> | "A man is walking through corn fields with a gun while birds fly in the sky." | "A cap gun is fired and a bird chirps."  |
> | "A toilet bowl with water in it."                                  | "Toilet flushing with water."                     |
>
> ---
>
> **[Q4] Can the model answer correctly even under text bias or counterfactual inputs?**
>
> To assess text bias, we evaluate the VideoLLaMA2 baseline with both modalities masked. The model consistently respond with "There is silence", demonstrating that it does not generate stereotypical answers solely from the textual prompt "Describe what you see and hear in a single sentence.". This observation is further corroborated by our findings in **[Q2]**, where single-modality masking cause the model to describe only the remaining modality (e.g., masking audio led to purely visual descriptions). Together, these results confirm that the model exhibits strong grounding capabilities and is not driven by textual priors. Consequently, we considered text bias to be a minimal factor and concentrated our study on the dominance issue between audio and video, where we identified significant room for improvement.
>
> Regarding counterfactual or inconsistent inputs, we analyze the behavior of VideoLLaMA2 on the synthetic (mismatched) pairs of the AVHBench AV Matching subset. We observe a bias in the baseline: it incorrectly predict "Yes" in about 59.53% of these inconsistent cases, likely driven by a textual prior assuming audio-visual correspondence. However, applying FMD effectively mitigate this issue. Our method correctly shift these predictions to "No", resulting in an 6.75% performance improvement on these negative pairs.
> This evidence suggests that FMD not only balances audio and visual modalities but also indirectly alleviates overreliance on textual priors, forcing the model to ground its answers in the actual sensory content.
>
> ---

---

### Official Review · Reviewer_17oF · 2025-10-31

**Soundness:** 2
**Presentation:** 3
**Contribution:** 3
**Rating:** 4
**Confidence:** 4

**Summary:**

This paper focuses on the issue of modality bias in current audio-visual large language models (AV-LLMs), where models tend to over-rely on the visual modality while overlooking the audio. To mitigate this, the authors propose Fork-Merge Decoding (FMD), an inference-time strategy that works for both token-wise fusion and channel-wise fusion AV-LLMs. The decoding process is divided into two stages:
  -  **fork**: the model performs modality-specific reasoning by masking out the other modality
  - **merge**: the resulting hidden states are combined for joint multimodal reasoning

The method is evaluated on three representative AV-LLMs, achieving consistent improvements over vanilla decoding strategies and other inference-time contrastive decoding strategies.

**Strengths:**

1. **Novel yet simple method:** FMD is easy to implement, making it broadly applicable across different AV-LLMs.
2. **Comprehensive analysis:** detailed quantitative studies, including attention analysis and ablation experiments on fork layers are conducted
3. **Latency:** FMD sets the fork layer to be early layers. This leads to limited additional computation, and may be optimized by engineering (e.g., parallel processing masked audio and video inputs)

**Weaknesses:**

1. **Dataset/task-specific $\alpha$:** The optimization of $\alpha$ for specific datasets limits the robustness of FMD. For real-world applications, it may be infeasible to retrieve representative samples in advance.
2. **Models and benchmarks for validation:** Although representative AV-LLMs have been evaluated on benchmarks, the robustness still needs to be assessed on more models, such as reasoning models, since the improvement on VideoLLaMA2 and Qwen2.5-Omni is slight

**Questions:**

1. More base models and benchmarks. Suggestions: model - Video-SALMONN 2; benchmarks - Video-MME, WorldSense, AVUT
2. L235-236: "We do not apply attention-guided fusion in the channel-wise setting, as the hidden states do not disentangle audio and visual embeddings." This is confusing as audio and video features are concatenated along the channel axis. Do you mean audio and visual embeddings are not disentangled along the time axis?
3. I have concerns about 100 samples for calibration. Do they cover main domains? What is the influence of sample numbers on the performance?

---

> ### Author Response · Authors · 2025-11-21
>
> Thanks to the reviewer’s insightful comments, we were able to add detailed information that enhances reader understanding. For your convenience, all revisions and additions have been clearly highlighted in purple in the updated PDF. Based on the comments, we have:
>
> > **(1)** added experiments demonstrating the robustness of α across different sample sizes,
> > **(2)** evaluated a more recent model and additional benchmarks to further verify the effectiveness of FMD,
> > **(3)** clarified the statements in L235–236.
>
> Our detailed responses are provided below.
>
> ---
>
> **[W1, Q3] Concern about the robustness of α and its sensitivity to the number of samples used to estimate it.**
>
> We acknowledge that α is estimated using 100 samples in advance, as discussed in our limitation section. However, although α is selected using only AVHBench, it generalizes well across different datasets, including AVUT and WorldSense (these results are reported in our response to **[W2]**).
>
> Importantly, estimating α requires minimal effort. To demonstrate this, we pool samples from three datasets (AVHBench, AVUT, and WorldSense) and randomly sampled small batches (10 or 50 examples) from this unified set, repeating the process 100 times using the VideoLLaMA2 model.
> As shown in the table below, the resulting α values remain stable, with small variance. Since α is rounded to a coarse value such as 0.8, these fluctuations have no practical impact. This demonstrates that our α selection is lightweight, robust, and not tied to any specific tasks.
>
> | **Number of Samples** | **10**     | **50**     | **100**    |
> |-------------------|:--------:|:--------:|:--------:|
> | Mean              | 0.829  | 0.829  | 0.829  |
> | Std               | 0.024  | 0.010  | 0.007  |
>
> ---
>
> **[W2, Q1] Requests for more models and benchmarks.**
>
> To validate the robustness of FMD, we apply it to the latest AV-LLM, video-SALMONN 2+ **[1]**, as suggested. Experiments are conducted on both AVHBench(A→V, V→A, and AV matching), and WorldSense datasets. As shown in the table below, FMD consistently improves the performance of this recent AV-LLM across all benchmarks.
>
> | **Model**         | **AVHBench (Vanilla / FMD)** | **WorldSense (Vanilla / FMD)** |
> |-------------------|:----------------------------:|:-------------------------------:|
> | video-SALMONN 2+  | 65.07 / **65.70**            | 46.43 / **46.81**               |
>
> Furthermore, to verify the generalizability of our method, we extend our evaluation to two additional benchmarks, AVUT **[2]** and WorldSense **[3]**, using the same models as in the main paper (VideoLLaMA2 and video-SALMONN). As shown in the table below, FMD yields consistent improvements on these datasets, underscoring its robustness.
>
> | **Model**       | **AVUT (Vanilla / FMD)** | **WorldSense (Vanilla / FMD)** |
> |-----------------|:-------------------------:|:-------------------------------:|
> | VideoLLaMA2     | 44.44 / **45.13**         | 29.41 / **29.91**               |
> | video-SALMONN   | 32.36 / **32.85**         | 56.20 / **57.83**               |
>
> > **[Reference]**
> > **[1]** *Video-SALMONN 2: Caption-Enhanced Audio-Visual Large Language Models, arXiv 2025.*
> > **[2]** *Audio-centric Video Understanding Benchmark without Text Shortcut, EMNLP 2025.*
> > **[3]** *WorldSense: Evaluating Real-world Omnimodal Understanding for Multimodal LLMs, arXiv 2025.*
>
> ---
>
> **[Q2] Concern about the confusing explanation in L235–236.**
>
> The reviewer is correct. In the channel-wise setting, modalities are fused within the channel dimension and thus not disentangled along the time axis. This prevents us from isolating the contribution of each modality via attention scores. We will clarify this phrasing in the final version.
>
> ---

---

> ### Comment · Reviewer_17oF · 2025-11-24
>
> Thanks for the effort of involving more recent AV-LLMs and audio-visual understanding benchmarks. The overall results show that FMD shows consistent improvements for different models, on different benchmarks. The improvement becomes quite marginal when the base AV-LLM becomes stronger (e.g.,Qwen2.5-Omni and video-SALMONN 2+). Due to the consistent improvement, I am willing to raise my score.
>
> However, the paper has not been updated. Please update the paper based on additional results and analysis.

---

> > ### Author Response · Authors · 2025-11-25
> >
> > Dear Reviewer 17oF,
> >
> > We sincerely thank you for your constructive and thorough feedback. We have fully updated our paper and incorporated all requested results and analyses. For your convenience, all revisions and additions are clearly highlighted in purple text.
> >
> > Sincerely,
> > The Authors.

---

### Official Review · Reviewer_ZLb5 · 2025-11-02

**Soundness:** 2
**Presentation:** 3
**Contribution:** 2
**Rating:** 2
**Confidence:** 4

**Summary:**

This paper proposed Fork-Merge Decoding (FMD) inference-time strategy: first, audio-only and video-only inputs are processed separately through the early layers of the decoder (the fork phase), and then the hidden states are merged to perform joint reasoning through the remaining layers (the merge phase). This method has been validated on three representative AV-LLMs (VideoLLaMA2, video-SALMONN, and Qwen2.5-Omni) and three benchmark datasets (AVQA, MUSIC-AVQA, and AVHBench), achieving consistent performance improvements in audio, video, and audio-visual joint reasoning tasks.

**Strengths:**

- The design of FMD avoids additional training or architectural modifications.
- The paper is easy to follow.

**Weaknesses:**

- Lack of technical novelty: This method does not alter the fundamental nature of modality imbalance. During training, visual information has higher information density, and there are cases where the model can answer correctly by focusing solely on one modality—these factors lead to the model assigning different levels of attention to different modalities. Therefore, even though the model adopts a separate masking strategy, it still fails to address the aforementioned issues and thus cannot effectively resolve the modality imbalance problem.
- The modality masking method is relatively suitable for samples with consistent audio and video. For example, in the cases provided in Figure 2, the audio and video are completely consistent: either the model is required to output descriptions that include both audio and video content (case a), or the audio provides supplementary information for the video (e.g., the video fails to clarify the type of vehicle, but the audio contains the sound of a train, which helps the model make the correct selection, case b). However, this method cannot handle scenarios where audio and video are inconsistent. For instance, when asked about a sound-producing object that does not appear in the video, masking the audio and relying solely on the video is unnecessary.
- There is a feature space gap between audio features and video features. Therefore, directly fusing them in the merge phase is unreasonable, as it fails to account for this gap between the features.
- The method lacks performance analysis on unimodal understanding of audio and video. According to the modality masking strategy proposed by the authors, it seems impossible to achieve performance improvement in the understanding of video-only or audio-only content.
- This training-free approach cannot enhance the model’s inherent performance on fine-grained tasks, such as temporal grounding and spatial grounding. The authors do not conduct relevant evaluations in the paper.
- How does the method perform on unimodal benchmarks for video or audio? Examples include video-only benchmarks like Video-MME, Video-MMMU, MMVU, MVBench, MMBench-Video, LongVideoBench, EgoSchema, PerceptionTest, MLVU, LVBench, TempCompass, and Charades-STA, as well as other benchmarks for audio/music understanding
- No performance evaluation on unimodal (video-only/audio-only) benchmarks: The article only tests FMD on audio-visual joint benchmarks (AVQA, MUSIC-AVQA, AVHBench) and completely ignores unimodal benchmarks—including video-only benchmarks (e.g., Video-MME, MVBench, MMBench-Video, LongVideoBench) and audio-only benchmarks.

**Questions:**

See weaknesses.

---

> ### Author Response · Authors · 2025-11-21
>
> We appreciate the reviewer’s helpful feedback. All revisions are highlighted in purple in the updated PDF. In response, we have:
>
> > **(1)** re-emphasized that FMD contributes to achieving balanced multimodal understanding,
> > **(2)** referenced results for audio–visual inconsistent cases,
> > **(3)** explained our merge process,
> > **(4)** clarified the scope of FMD,
> > **(5)** added experiments verifying robustness on fine-grained tasks.
>
> Our detailed responses are provided below.
>
> ---
>
> **[W1] Can the proposed method mitigate the modality-imbalance problem, as indicated by the uneven distribution of attention weights?**
>
> **Figure 1** in the main paper shows that pretrained AV-LLMs exhibit a strong bias towards video tokens over audio cues. Our FMD addresses this imbalance by lowering the attention given to video tokens and elevating the attention to audio tokens. Importantly, FMD preserves the model’s original attributes (i.e. video attention still remains higher than audio attention), but allows the model to better exploit the less utilized modality. Consequently, FMD offers a simple, plug-in decoding method that enhances balanced multimodal understanding.
>
> ----
>
> **[W2] Can the method handle scenarios where the audio and video are inconsistent?**
>
> Contrary to the reviewer’s concern, our method also improves performance in audio–visual inconsistent cases. The AV matching subset of AVHBench, which we evaluate on, explicitly includes both matched pairs and synthetically mismatched or misaligned audio-video pairs, which are exactly the inconsistent settings the reviewer refers to. Even under these conditions, FMD yields substantial gains (e.g., more than 5% absolute improvement on video-SALMONN), as shown in **Table 1**, indicating that the model no longer over-relies on a single modality.
>
> Moreover, AVHBench includes audio-driven video hallucination cases (A→V), which directly correspond to situations such as “a sound-producing object not appearing in the video.” FMD improves performance in these cases as well (**Table 1**).
>
> This demonstrates that FMD remains robust even when one modality is misleading or irrelevant.
>
> ---
>
> **[W3] Is direct fusion appropriate when the audio and video features reside in mismatched feature spaces that are not aligned?**
>
> Our method does not directly fuse unaligned audio/video embeddings. First, all models used in our experiments already include a learned mapping network that aligns audio and video features into a shared LLM input space, and our method operates only after this pretrained alignment. Second, in the merge stage: (1) video tokens are merged only with video tokens, (2) audio tokens are merged only with audio tokens. This means we never perform cross-modal fusion during this step. Therefore, the reviewer’s concern does not apply, as our method preserves the modality-specific feature structure of the pretrained model.
>
> ---
>
> **[W4, 6, 7] A concern is raised regarding the absence of unimodal evaluation.**
>
> Our work focuses on improving the inference-time behavior of AV-LLMs. FMD operates by running (1) (audio+prompt) and (2) (video+prompt) inference separately in the fork stage to strengthen each modality’s representation, followed by aggregation in the merge stage (video+audio+prompt). This process fundamentally requires two distinct modalities except for the prompt. In unimodal settings (video-only or audio-only), there is no second modality to isolate, enhance, or rebalance, so the procedure cannot be instantiated. Therefore, unimodal-only benchmarks fall outside the scope of our method and are not the intended application setting for AV-LLMs.
>
> ---
>
> **[W5] Can the method improve performance on fine-grained tasks?**
>
> We appreciate the reviewer’s suggestion. To evaluate whether our training-free method enhances fine-grained reasoning, we test FMD on the AVUT **[1]** and WorldSense **[2]** datasets, which are specifically designed for fine-grained audio and video understanding, including temporal and spatial grounding. Due to VideoLlama2's sparse 8-frame sampling, we filter videos to $\le$1 minute.
> Using VideoLLaMA2 and video-SALMONN, our method consistently improves performance on both benchmarks, which require careful temporal and spatial reasoning. These results demonstrate that even tasks demanding fine-grained temporal understanding benefit from more balanced multimodal utilization at test time.
>
> | Dataset     | VideoLLaMA2 (Vanilla / **FMD**) | video-SALMONN (Vanilla / **FMD**) |
> |-------------|:----------------------------:|:------------------------------:|
> | AVUT        | 44.44 / **45.13**                | 32.36 / **32.85**                  |
> | WorldSense  | 29.41 / **29.91**                | 56.20 / **57.83**                  |
>
> > **[Reference]**
> > **[1]** *Audio-centric Video Understanding Benchmark without Text Shortcut, EMNLP 2025.*
> > **[2]** *WorldSense: Evaluating Real-world Omnimodal Understanding for Multimodal LLMs, arXiv 2025.*
>
> ---

---

### Meta-Review · Area_Chair_Z1XC · 2025-12-19

**Summary:**

The AC carefully read the paper and all discussions. This paper received mixed reviews (initial scores of 2, 4, 4, 4, and 6, with one reviewer later increasing their score from 4 to positive based on the discussion). While reviewers generally acknowledged the simplicity of the proposed method and some positive empirical results, the main concerns relate to the marginal performance gains and the sensitivity to hyperparameter choices, which limit the method’s robustness, generality, and practical usability. Overall, it remains unclear whether the proposed approach can provide clear or meaningful benefits for current or future models. The aggregated reviewer scores lean toward rejection, and given the inherent limitations, it appears difficult to shift the overall consensus through discussion. Therefore, I am inclined to recommend rejection.

**Reviewer Concerns:**

Some concerns regarding the evaluation scenarios and more recent model (reviewer 17oF, wsv4, 4GtS), and the choice of some of the hyperparameters (reviewer 17oF, xuKV) have been addressed. However, non-negligible concerns still remain:

1. The performance gains remain marginal (also noted by reviewers 17oF and wsv4), especially given that the method also introduces additional inference cost (reviewer xuKV, though lower than some existing approaches). Moreover, the improvements appear to diminish as the base model becomes stronger, as also mentioned by reviewer 17oF.

2. The method relies on several hyperparameter choices, some of which are relatively sensitive (e.g., layer selection; as noted by reviewers xuKv and wsv4), raising concerns about robustness, generality, and practical usability.

Besides, reviewer ZLb5’s concerns regarding novelty and the scope of applicability of the method are difficult to fully address, even though such assessments may involve subjectivity and can vary across different reviewers.

Overall, it remains unclear whether the proposed approach can provide clear or meaningful benefits for current or future models.

**Reviewer Scores:**

1. Reviewer ZLb5 may at most slightly increase the score, but would still lean toward rejection, mainly due to concerns about the novelty and the scope of applicability.
2. Reviewer 17oF raised the score based on the context of the discussion.
3. Reviewer xuKv is likely to maintain the original score or make only a minor adjustment. The reviewer did not seem to exhibit misunderstandings of the paper. While the authors addressed the raised concerns, the method itself still has inherent limitations (e.g., inference speed and sensitivity to hyperparameters).
4. Reviewer wsv4 will likely keep the original score, as the method indeed suffers from inherent limitations such as marginal performance improvements and sensitivity to hyperparameters.
5. Reviewer 4GtS is also likely to maintain the original score. This reviewer already gave a positive rating, and the issues raised, together with the authors’ responses, do not seem sufficient to justify a further score increase.

---

### Decision · Program_Chairs · 2026-01-26

Reject